# SpokenWOZ: A Large-Scale Speech-Text Benchmark for Spoken Task-Oriented Dialogue Agents

**Shuzheng Si**[1]\*, **Wentao Ma**[1]\*, **Haoyu Gao**[1]\*, **Yuchuan Wu**[1], **Ting-En Lin**[1],
**Yinpei Dai**[2]†, **Hangyu Li**[1], **Rui Yan**[3]‡, **Fei Huang**[1] **and Yongbin Li**[1]‡
[1]DAMO Academy, Alibaba Group
[2]Computer Science and Engineering Division, University of Michigan
[3]Gaoling School of Artificial Intelligence, Renmin University of China
sishuzheng@foxmail.com, {mawentao.mwt, ghy385779}@alibaba-inc.com
https://spokenwoz.github.io/SpokenWOZ-github.io/

## Abstract

Task-oriented dialogue (TOD) models have made significant progress in recent years. However, previous studies primarily focus on datasets written by annotators, which has resulted in a gap between academic research and real-world spoken conversation scenarios. While several small-scale spoken TOD datasets are proposed to address robustness issues such as ASR errors, they ignore the unique challenges in spoken conversation. To tackle the limitations, we introduce **SpokenWOZ**, a large-scale speech-text dataset for spoken TOD, containing **8** domains, **203k** turns, **5.7k** dialogues and **249 hours** of audios from human-to-human spoken conversations. SpokenWOZ further incorporates common spoken characteristics such as word-by-word processing and reasoning in spoken language. Based on these characteristics, we present cross-turn slot and reasoning slot detection as new challenges. We conduct experiments on various baselines, including text-modal models, newly proposed dual-modal models, and LLMs, e.g., ChatGPT. The results show that the current models still have substantial room for improvement in spoken conversation, where the most advanced dialogue state tracker only achieves 25.65% in joint goal accuracy and the SOTA end-to-end model only correctly completes the user request in 52.1% of dialogues. Our dataset, code, and leaderboard are available at https://spokenwoz.github.io/SpokenWOZ-github.io/.

## 1 Introduction

Dialogue system modeling, a classic research topic in the field of human-machine interaction, serves as an important application area [22, 21, 35, 44, 36, 34]. Recently, task-oriented dialogue (TOD) systems [14, 41, 3] have attracted significant attention. These systems are designed to assist users in accomplishing specific goals, e.g., flight booking and restaurant reservation. Benefited from the previous well-designed data-collection pipeline, such as Wizard-of-Oz [19], the TOD datasets [39, 2] can be constructed with low costs by writing from annotators, rather than being collected from realistic spoken conversations.

However, these TOD datasets constructed solely based on written texts may not accurately reflect the nuances of spoken conversations, leading to a gap between academic research and real-world spoken TOD scenarios. To advance the research on more realistic spoken TOD, various datasets have been

---

\*Equal contribution.

†Work done while the author was working at Alibaba.

‡Corresponding author.

37th Conference on Neural Information Processing Systems (NeurIPS 2023) Track on Datasets and Benchmarks.

Table 1: Dataset statistics of SpokenWOZ and existing TOD datasets *: SpokenWOZ contains 4200 dialogues in the training set.

| Metric | DSTC2 | KVRET | M2M | MultiWOZ | ABCD | DSTC10 | SpokenWOZ* |
|---|---|---|---|---|---|---|---|
| **Dialogues** | 1,612 | 2,425 | 1,500 | **8,438** | 8,034 | 107 | 5,700 |
| **Turns** | 23,354 | 12,732 | 14,796 | 115,424 | 177,407 | 2,292 | **203,074** |
| **Domains** | Single | **Multi** | **Multi** | **Multi** | **Multi** | **Multi** | **Multi** |
| **Collection** | H2M | **H2H** | M2M | **H2H** | **H2H** | **H2H** | **H2H** |
| **Type** | **Spoken** | Written | Written | Written | Written | **Spoken** | **Spoken** |
| **Audio** | ✅ | ❌ | ❌ | ❌ | ❌ | ❌ | ✅ |
| **Cross-turn Slot** | ❌ | ❌ | ❌ | ❌ | ❌ | ❌ | ✅ |
| **Reasoning Slot** | ❌ | ❌ | ❌ | ❌ | ❌ | ❌ | ✅ |

introduced to simulate and model spoken conversations [16, 17, 20]. Despite these efforts, existing datasets still exhibit three notable weaknesses:

• **Data Scale**: Previous spoken TOD datasets have been limited in terms of data scale, posing challenges for developing strong and more realistic systems. The pioneering ATIS dataset [16] consists of only 41 dialogues, which severely restricts the diversity and breadth of dialogues. While recent efforts, such as DSTC2 [17] and DSTC10 [20], have renewed interest in modeling spoken TOD, they offer a modest improvement with 1,612 and 107 dialogues, respectively. In contrast, written TOD datasets offer a significantly larger number of dialogues, exemplified by MultiWOZ [2]. Therefore, addressing this data scale limitation is crucial to advancing the spoken TOD systems.

• **Human-to-Human Audio**: Another limitation is the lack of audio from human-to-human spoken conversations. DSTC2 collects the spoken audio via the Human-to-Machine schema, limiting the construction of human-like systems. DSTC10 only provides the automatic speech recognition (ASR) hypotheses and dialogue text content, limiting the investigation of the dual-modal TOD model.

• **Unique Challenges in Spoken Conversations**: Previous spoken TOD datasets focus on robustness issues, especially ASR noise, ignoring the unique characteristics of spoken conversation, including word-by-word processing and reasoning in spoken language. These characteristics bring new challenges, such as handling incomplete utterances and reasoning in spoken languages.

Therefore, we aim at building a spoken TOD dataset from human-to-human spoken conversations, named SpokenWOZ, by extending the Wizard-of-Oz pipeline to realistic spoken conversations with real-time voice calls. To better model the spoken characteristics in SpokenWOZ, we introduce cross-turn slot and reasoning slot detection as new challenges to respectively address the dialogue state tracking in incomplete utterances and the indirect expression.

We spend more than 8 months building SpokenWOZ and implementing various quality improvements, achieving the final turn-level annotation accuracy of over 97%. The total cost of constructing SpokenWOZ is approximate $55k, including $30k for audio collection, $20k for dialogue annotation, and $5k for the cost of using ASR tools.

In summary, our contributions are threefold:

- We introduce a large-scale speech-text TOD dataset named SpokenWOZ, which contains more than 203K annotated utterances, 5,700 dialogues, and the associated 249 hours of audio. To the best of our knowledge, this is the largest speech-text TOD dataset with annotation of the dialogue state and dialogue act.
- We identify the new challenges in spoken conversation based on a comprehensive analysis of SpokenWOZ, including cross-turn slot detection and reasoning slot detection, which are not taken into account by the previous TOD datasets.
- We also conduct comprehensive experiments on various baselines, including text-modal TOD baselines and newly proposed dual-modal models. We also explore the LLM's zero-shot performance on SpokenWOZ, e.g., ChatGPT. The results demonstrate the models still have much room for improvement in spoken conversation scenarios.

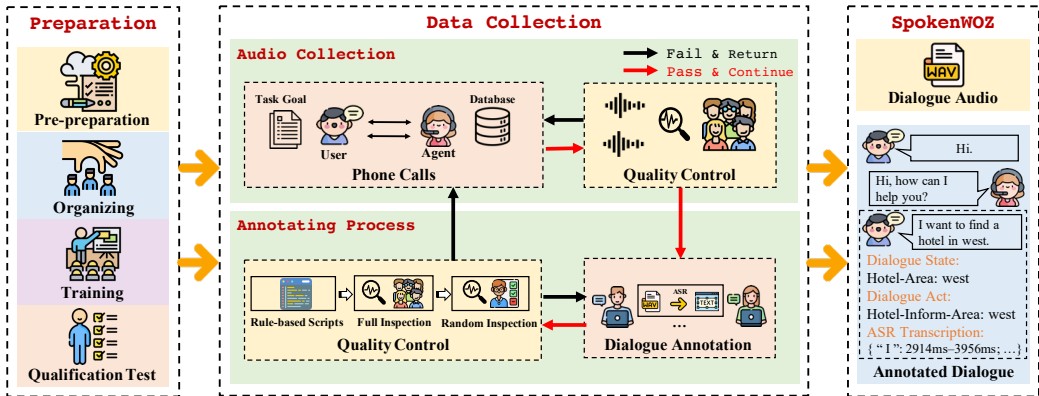

Figure 1: Construction schema of SpokenWOZ. Data collection includes (1) collection of dialogue audio and (2) annotation of dialogue. Strict quality control is performed at each collection stage.

## 2 Related Work

**Written TOD datasets** To push forward research in TOD modeling, various written TOD datasets have been proposed, ranging from human-to-machine [31] to more natural human-to-human dialogues [2], as well as from single-domain [39] to more realistic multi-domain scenarios [29]. Notable written TOD datasets include M2M [33], KVRET [11], MultiWOZ [2], SGD [30], and ABCD [4]. Initially, these datasets focus on the single-domain dialogue. Subsequently, MultiWOZ and SGD are proposed to address more realistic multi-domain dialogue. Recently, numerous written TOD datasets attempt to simulate more realistic spoken conversations. For instance, RADDLE [27] manually introduces various human-designed spoken noises to the written MultiWOZ. CGoDial [6] adds spoken features to existing Chinese TOD datasets via crowd-sourcing, but the language and lack of audio data both limit the research on spoken TOD. Similarly, SSTOD [43] introduces segment-by-segment interactions in simulated spoken language. Nonetheless, these studies concentrate solely on specific aspects of spoken TOD, thereby restricting a more holistic examination.

**Spoken TOD datasets** Several datasets have been proposed for modeling the spoken TOD. ATIS [16] focuses on single-domain TOD and only contains 41 dialogues. DSTC2 [17] provides spoken corpora, but remains limited in scale. DSTC10 [20] revisits spoken TOD, providing 107 dialogues that include ASR hypotheses. EVI [37] provides a transcript and ASR hypotheses, but only evaluates limited tasks, including enrolment (E), verification (V) and identification (I), ignoring more common tasks in TOD, such as querying and booking. To the best of our knowledge, SpokenWOZ represents the first large-scale speech-text dataset for TOD with annotation of the dialogue state and act.

## 3 SpokenWOZ Construction

To ensure the reliability of SpokenWOZ, we will discuss the stages involved in the construction schema in Figure 1. As shown in Table 2, SpokenWOZ consists of 8 different domains, 7 of which are inherited from widely used MultiWOZ, reducing the usage burden. Meanwhile, we design a new domain "profile" to introduce a realistic scenario where the agent needs to collect the user's personal information as the booking confirmation.

### 3.1 Collection of Dialogue Audio

To obtain dialogue audio in realistic spoken conversation, we organize 250 participants to generate 5,700 dialogues via phone calls. During the collection, one participant plays the role of a user and asks questions based on a template-generated task goal as shown in Appendix A.7. The other participant plays the role of an agent and answers the user by searching the same database as MultiWOZ. We build an online database for the agent as shown in Appendix A.8 to enhance the realism of the dialogues. More details about dialogue audio can be found in Appendix A.1.4.

Table 2: Full ontology in SpokenWOZ. The upper script indicates which domains it belongs to. *: universal, 1: restaurant, 2: hotel, 3: attraction, 4: taxi, 5: train, 6: hospital, 7: police, 8: profile.

| | |
|---|---|
| ***acts*** | inform* / request* / ack* /select$^{123}$ / recommend$^{123}$ / nooffer$^{123}$ / nobook$^{125}$ / book$^{125}$ edit$^8$ / confirm$^8$ / greet / bye / reqmore / wait / backchannel / thanks |
| ***slots*** | address$^{12367}$ / postcode $^{12367}$ / phone $^{123678}$ / name$^{12348}$ / area$^{123}$ / price range$^{123}$ type$^{123}$ / internet$^2$ parking$^2$ / stars$^2$ / open hours$^3$ / departure$^{45}$ / destination$^{45}$ leaveat$^{45}$ / arriveby$^{45}$ / ID$^8$ / email$^8$ / license plate number$^8$ / bookpeople$^{125}$ trainID$^5$ / ticket price$^5$ / booktime$^1$ / department$^7$ / bookday$^{12}$ / day$^5$ / bookstay$^{12}$ |

**Qualification Test.** Before the collection, we conduct a qualification test for each participant to ensure the quality of the data. The criteria include whether the task goal is completed and the naturalness of the audio. Each group of participants should submit completed dialogue audio, and we judge whether they pass the test. We initially receive 1520 applications, of which only 250 (16.4%) pass the qualification test.

**Quality Control.** We employ crowd-sourcing to evaluate the quality of each audio and remove the audio that exhibit poor communication quality or did not fulfill the task goal.

**Speaker Origins.** To ensure dialogue diversity, we organize participants from various countries, including Canada, Singapore, China, and South Africa. More details are shown in Appendix A.1.3.

## 3.2 Annotation of Dialogue

We train 15 annotators to annotate the clean dialogue state and dialogue act using both text transcriptions generated by the ASR tool [4] and audio files. We inherit and expand the annotation schema of MultiWOZ, e.g., adding the new act "backchannel" to better model spoken conversations. Meanwhile, annotators need to transcribe the agent utterances according to the audio to obtain a clean text. In order to keep the noise from ASR tool, we do not manually transcribe the user utterances and keep the ASR noise in the user utterances. Due to the finely adequate training (approximately 3 weeks), this annotating process was able to achieve better results.

**Qualification Test.** We conduct a qualification test for each annotator. The test is only passed if the test dialogues are annotated completely correct by annotators. This approach ensures that only qualified annotators could participate in the final annotation to improve the quality of SpokenWOZ.

**Quality Control.** Before annotation, the annotators will return the audio in low quality. To ensure the quality of the annotations, we implement a three-step quality control process, including script checking based on rules, full inspection by annotators, and random inspection by us. We use scripts to identify simple errors such as missing annotations and return the dialogues for revision. Next, annotators conduct a full inspection and correct any wrong labels. Finally, we carry out a random inspection on 10% of the dialogues in SpokenWOZ. If the correct rate of the random inspection was less than 97% at turn-level, we would return this batch of dialogues for re-inspection by the annotators. Due to the strict quality control, SpokenWOZ can achieve a much higher quality than MultiWOZ, which contains over 40% of dialogue turns that need to be modified [10].

## 4 SpokenWOZ Dialogue Corpus

We build a novel, large-scale, human-to-human, multi-domain, dual-modal dataset that can be used to develop TOD systems on spoken conversations. SpokenWOZ includes a total of 5,700 dialogues and more than 203k utterances along with 249 hours of audio.

### 4.1 Characteristics of SpokenWOZ

In this section, we will highlight the unique spoken characteristics in SpokenWOZ, including word-by-word processing, ASR noise, and reasoning in spoken language. We will use the dialogue segments

---

[4]https://www.alibabacloud.com/help/en/intelligent-speech-interaction/latest/recording-file-recognition

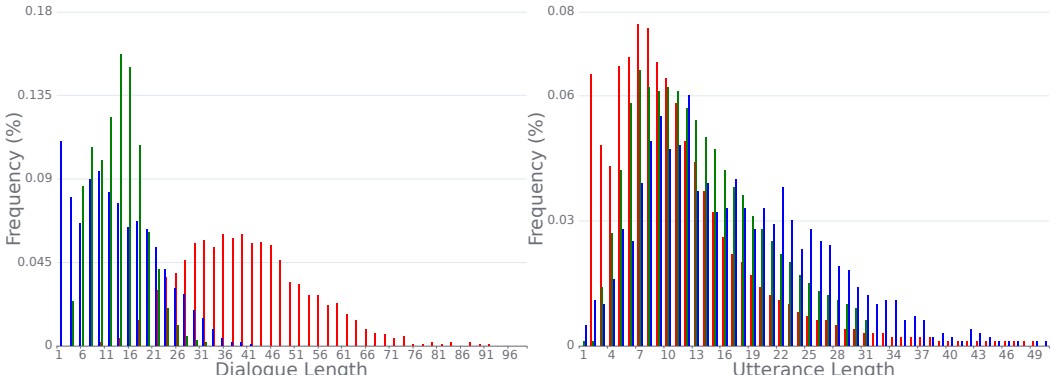

Figure 2: The distribution of SpokenWOZ, written MultiWOZ and spoken DSTC10. The first two datasets have similar domains, and we can observe that spoken TOD tends to have more turns but shorter utterances. Compared to DSTC10, SpokenWOZ has more turns due to the cross-turn slots, and has shorter utterances because it does not model the knowledge-grounded TOD as DSTC10 does.

from SpokenWOZ to illustrate these characteristics in this section. Understanding these complexities is essential for developing more realistic spoken TOD systems. Additionally, SpokenWOZ has a distinct distribution compared with the written MultiWOZ, as shown in Figure 2, demonstrating the difference between written and spoken TOD. More statistics are illustrated in Appendix A.5.

### 4.1.1 Word-by-Word Processing

Most TOD datasets only include written conversations collected via crowd-sourcing. As a result, the interactions are turn-by-turn, with each turn containing complete semantics, such as *"I would like a cheap restaurant in the north area."*. However, language processing in spoken conversation is inherently incremental and word-by-word. The current word spoken is not necessarily well thought out and relies more on the preceding few words, such as *"I would like a restaurant, hmmm, cheap one please, meanwhile in the south area, oh, sorry, in the north."*. This leads to less organized interactions, with utterances often characterized by various linguistic phenomena such as back-channel, disfluencies, and incomplete utterances.

**Back-channel.** Back-channel is a common phenomenon in human-to-human interaction. To effectively communicate with others, interlocutors often use minor messages such as "hmm" to show their cooperation and engagement during a conversation. This can be quite challenging when one participant responds without enough semantics to follow the previous statement.

**Disfluencies.** Another spoken language phenomenon is disfluencies, such as repeating and pausing, which introduce noise in utterance text. The disfluencies in spoken language introduce noise, leading to increased demands on model robustness.

**Incomplete Utterances.** In spoken conversations, users tend to inform long information in multiple utterances rather than conveying it all in one utterance. The information is provided segment-by-segment over multi-turn interactions. For example, the user can tell the license plate in the following utterances, *"Okay. My license plate is n e"*, *"8 6"* and *"g z w"*. In written TOD datasets, one complete utterance can contain all information, like *"my license plate number is ne86gzw"*. How to integrate the information distributed in multiple turns is a new challenge.

### 4.1.2 ASR Noise

In spoken dialogue systems, the ASR module is integrated to convert speech into text, but it also introduces noise. Previous approaches, like RADDLE, attempt to simulate ASR errors by manually writing noisy texts to the written TOD datasets. In contrast, SpokenWOZ takes a different approach by naturally introducing ASR noise from real-world scenarios. This is achieved by first obtaining

dialogue audio and then using an ASR module to transcribe the audio into text. The presence of ASR noise can pose challenges for the system to understand the user's intent and generate a response.

### 4.1.3 Reasoning in Spoken Language

Given the casual and informal nature of spoken language, accurately determining the user's intent may require reasoning. For instance, Table 3 in SpokenWOZ demonstrates a scenario where the model needs to conduct reasoning to determine the total number of people involved.

Table 3: The dialogue segment from SpokenWOZ shows the reasoning in spoken language.

---

: Yeah. Yeah. Uh, can you book it for **me, my parents and my grandparents?**

: Okay, so it's **five** people in total.

---

## 4.2 New Challenges of SpokenWOZ

The inherent traits of spoken language pose unique challenges to TOD systems. To address this, we introduce two new types of slots in SpokenWOZ, including cross-turn slots and reasoning slots, for incomplete utterances and indirect expression in spoken language, respectively.

### 4.2.1 Cross-turn Slot Detection

In spoken conversations, users may provide the value of a slot across multiple turns rather than in a single turn. Each turn only provides a piece of the value, and this is called the cross-turn slot. We introduce 5 cross-turn slots, including phone number, id number, user name, email, and license plate number in the domain "profile". More details are listed in Appendix A.1. Table 4 demonstrates an example where the user provides id number segments in different turns sequentially. Differing from updating a whole

Table 4: The dialogue segment from SpokenWOZ shows the Cross-turn Slot Detection.

---

: Oh, my id number is **5 2 5** 8 ( *"8" is missed by the ASR tool, but appears in the audio*).
(Dialog State: id_number = 5258)

: So it's 5 2 5 8.

: Yes. and then **5 7 6 3**.
(Dialog State: id_number = 52585763)

: 5 7 6 3.

: And then **7 5 2 5**.
(Dialog State: id_number = 525857637525)

: 7 5 2 5.

: I'm sorry, **7 5 to 4**.
(Dialog State: id_number = 525857637524)

: Yes. okay, so it's 7524.

: And then **double 9 0 3**.
(Dialog State: id_number = 5258576375249903)

---

slot value in one turn, the agent needs to precisely locate the part of values that needs to be updated. Meanwhile, the agent should not only accumulate the slot value across the multiple turns but also make more fine-grained changes to the current slot value, such as correcting "525857637525" to "525857637524" by saying "I'm sorry, 7, 5 to 4".

### 4.2.2 Reasoning Slot Detection

To address the challenges presented by the casual nature of spoken language, TOD systems need to engage in reasoning to effectively complete slot updates. SpokenWOZ introduces three kinds of reasoning, including Temporal Reasoning [45], Mathematical Reasoning [26], and Semantic Reasoning [32]. More reasoning slot details can be found in Appendix A.1, including the name of the specific slot and how to construct the reasoning slots in dialogue.

Table 5: The dialogue segment from SpokenWOZ shows Temporal Reasoning.

---

: Uh, yes, and on which day please?

: Oh. yeah. And I think **today is Friday**, right. Uh we will be there **tomorrow**.
(Dialog State: Bookday = Saturday)

---

**Temporal Reasoning.** Understanding temporal information is crucial for understanding user intent. Such as properly understanding the date of arrival at the hotel as shown in Table 5.

**Mathematical Reasoning.** Once the user does not directly inform the number in the utterance, mathematical reasoning is required for the user's expression. As shown in Section 4.1.3 and Table 3, the agent needs to know from the user utterance that the value of the Bookpeople slot is 5.

**Semantic Reasoning.** To capture the intent from indirect expressions, the agent needs to perform semantic reasoning based on the available information. As shown in Table 6, since the user expresses a desire for sushi, it can be inferred that the purpose is to find a Japanese restaurant.

## 5 Tasks & Settings

The complexity and diverse spoken characteristics in SpokenWOZ make it a useful dataset for different TOD tasks, including dialogue state tracking and response generation.

**Dialogue State Tracking (DST).** As highlighted in Section 4, the challenges in DST involve understanding noisy utterances, managing cross-turn slots, and handling reasoning slots. We use the joint goal accuracy (JGA) [2] as the metric, which measures the ratio of turns for which the value of each slot is correct.

Table 6: The dialogue segment from SpokenWOZ shows Semantic Reasoning.

| |
|---|
| 😃: Yes, ma'am, the restaurant should serve **sushi** and should be in the center. (Dialog State: Restaurant-Type = Japanese; Restaurant-Area = centre) |
| 💻: Just to confirm that the restaurants are serving Japanese food in the centre. |
| 😃: That's correct, man. |

For response generation, the challenges are twofold:

**Policy Optimization.** System optimizes dialogue policy and generates the appropriate response based on user utterance and dialogue state.

**End-to-end Modeling.** System generates the correct response solely based on the user utterance, without explicit dialogue state information.

Following the previous TOD setting [2], we report the metrics of whether the system provides a correct entity (INFORM) and answers all the requested information (SUCCESS). A combined score (Comb) is computed by (INFORM+SUCCESS)×0.5+BLEU [25] as an overall quality measure.

## 6 Experiments

### 6.1 Baselines

We compare text-modal baselines, which include the following models:

**BERT+TripPy:** TripPy [15] uses BERT [7] as encoder and uses copy mechanisms to fill slots for DST, including span prediction, inform operations, and slot copy.

**UBAR:** UBAR [40] is acquired by fine-tuning the GPT-2 [28] on the sequence of the entire dialogue which is composed of user utterance, dialogue state, database result, system act, and system response.

**GALAXY:** GALAXY [13] is a pre-trained conversation model based on UniLM [9] that learns dialogue policy via semi-supervised learning. GALAXY keeps the same input and output format as UBAR to generate the dialogue state or system response.

**SPACE:** SPACE [12] is a pre-trained conversation model based on UniLM that takes into account both dialogue understanding and dialogue generation. SPACE keeps the same input and output format as UBAR to generate the dialogue state or system response.

**SPACE+TripPy:** We use SPACE to replace the original encoder BERT in TripPy as a new baseline.

We utilize WavLM [5] as the speech encoder and implement the following dual-modal models:

**SPACE+WavLM+TripPy:** We use a bi-encoder framework and then concatenate the representation outputs from SPACE and WavLM, then use Transformer [38] encoder as fusion layer to allow the interaction between the different modalities. We use the multi-modal outputs from the fusion layer as the representations for TripPy.

**SPACE+WavLM:** To utilize the generation ability of SPACE and the speech-modal information, we concatenate the user utterance embeddings from SPACE and user audio embeddings from WavLM as new user-side inputs and then generate the state or response by the SPACE decoder.

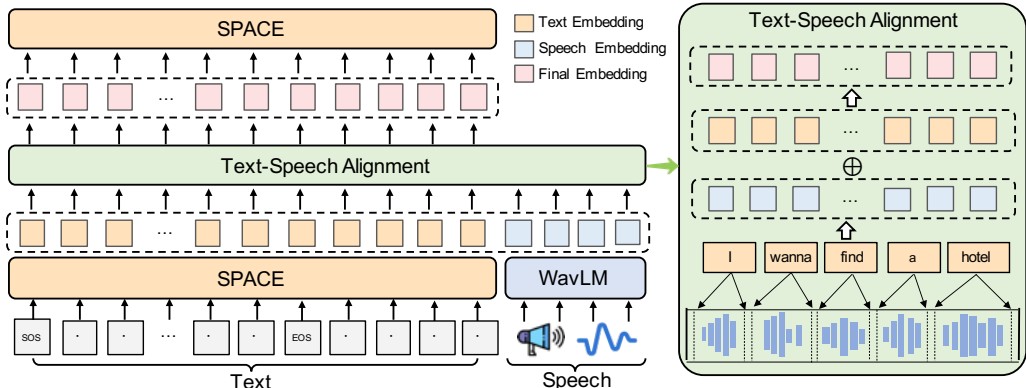

Figure 3: The architecture of SPACE+WavLM_align. It remains the same text-side inputs and formats as SPACE, i.e., sequence of the entire dialogue context, but additionally introduces aligned speech-modal information of user utterances.

**SPACE+WavLM$_{align}$:** As shown in Figure 3, we align the user-side word context and the corresponding audio pieces based on the annotations in SpokenWOZ. Then we add the user utterance embeddings from SPACE and the speech embeddings from WavLM as new user-side embeddings. During the inference, SPACE+WavLM$_{align}$ uses dual-modal input to generate the state and response.

As LLM makes amazing progress on NLP tasks [1], we also evaluate LLM's zero-shot performance. We use the prompts form Hudecek et al. [18] and Bang et al. [1] as reference.

**ChatGPT:** ChatGPT (GPT-3.5-turbo) is a conversational LLM [24], which has brought remarkable success on various zero-shot tasks.

**InstructGPT$_{003}$:** InstructGPT$_{003}$ (text-davinci-003) [24] is a 175B LLM trained by instruction tuning and reinforcement learning.

Please refer to Appendix A.2 for additional details and descriptions of the baseline models.

Table 7: DST experimental results.

| Model | JGA | -w/o cross-turn slot |
|---|---|---|
| BERT+TripPy | 14.78 | 15.58 |
| SPACE+TripPy | 16.24 | 17.31 |
| SPACE+WavLM+TripPy | 18.71 | 20.90 |
| UBAR | 20.54 | 23.51 |
| SPACE | 22.73 | 26.99 |
| SPACE+WavLM | 24.09 | 27.34 |
| **SPACE+WavLM$_{align}$** | **25.65** | **28.15** |
| ChatGPT | 13.75 | 16.30 |
| InstructGPT$_{003}$ | 14.15 | 16.49 |

## 6.2 Results and Discussion

The results are reported in Tables 7 and 8. These tables offer several key insights as follows:

**SpokenWOZ is challenging.** Compared with the results in written MultiWOZ, such as SPACE achieving a JGA of 57.5 in DST task and a Combined Score of 110.95 in End-to-end Modeling task, the metrics are significantly lower in SpokenWOZ. This indicates that the current models learn the written TOD well, but ignore the characteristics of spoken TOD. As shown in Table 7, all baselines get improved JGA performance without evaluating cross-turn slots, meaning modeling cross-turn slots is challenging for current models. Furthermore, we propose Macro Average Mentioned Slot Accuracy (MAMS Acc) to measure the difficulty of different types of slots in DST. MAMS Acc is calculated by (i) separately determining the accuracy of each slot, excluding instances where the " none" value is present in the final turn; (ii) in each slot category, we calculate the macro average for the accuracy of each slot belonging to this category to get the MAMS Acc. We divide all the slots into four categories: reasoning slot, cross-turn slot, ASR-sensitive slot, and normal slot. Then we select "name" in restaurant, hotel and attraction domains as ASR-sensitive slots, because the special entity names are difficult to correctly recognized via ASR tool. Normal slot includes slots that do not belong to the above three types. As shown in Figure 4, the spoken characteristic such as ASR noise, reasoning, cross-turn slots are quite challenging and there is still much room for progress in the research of spoken TOD model. In addition to the challenges in DST, response generation tasks face their unique challenge of more diverse act flow modeling. SpokenWOZ extends the ontology of

Table 8: Policy Optimization and End-to-end Modeling experimental results.

| Model | Policy Optimization | | | | End-to-end Modeling | | | |
|---|---|---|---|---|---|---|---|---|
| | INFORM | SUCCESS | BLEU | Comb | INFORM | SUCCESS | BLEU | Comb |
| UBAR | 62.50 | 48.10 | 9.69 | 64.99 | 60.20 | 47.40 | 9.90 | 63.70 |
| GALAXY | 70.60 | 42.20 | 16.52 | 72.92 | 65.80 | 38.50 | 20.10 | 72.25 |
| SPACE | 76.00 | 57.60 | 18.72 | 85.52 | 66.40 | 50.60 | 21.34 | 79.84 |
| SPACE+WavLM | 76.80 | 58.40 | 18.54 | 86.14 | 67.20 | 51.30 | 21.46 | 80.71 |
| **SPACE+WavLM$_{align}$** | 77.20 | **59.20** | **19.81** | **88.01** | **68.30** | **52.10** | **22.12** | **82.32** |
| ChatGPT | 73.40 | 39.50 | 4.58 | 61.03 | 23.40 | 13.80 | 3.59 | 22.19 |
| InstructGPT$_{003}$ | **78.20** | 56.90 | 7.72 | 75.27 | 25.30 | 18.50 | 6.13 | 28.03 |

MultiWOZ, which introduces the difficulties to predict the right system acts. Meanwhile, the act flow in SpokenWOZ is more diverse than in written MultiWOZ as shown in Appendix A.4.

**Dual-modal TOD models is what you need.** The significant size of the dialogues and audios in SpokenWOZ allows researchers to build large data-driven neural models given textual inputs and dual-modal inputs. To use dual-modal data in SpokenWOZ, we introduce several dual-modal baselines. Dual-modal methods achieve improved performance, showing the necessity of dual-modal methods in realistic scenarios. We observe that SPACE+WavLM$_{align}$ outperforms SPACE+WavLM due to the speech-text alignment. We show Case Study in Appendix A.3 to confirm our claim.

**Supervised generative methods are helpful.** We can empirically observe that generative methods, e.g., UBAR, achieve better performance than extractive methods, e.g., TripPy. We further conduct Case Study in Appendix A.3. It shows that the existing extractive methods can not handle cross-turn slots and reasoning slots, as the target values do not directly appear in the utterance. Meanwhile, due to the ASR noise, the extractive methods easily extract the wrong value, even in the right location. However, the generative methods can be robust to ASR noise and modify the wrong word in the original utterance to the correct value.

Figure 4: MAMS Acc of four categories of slot in advanced generative-methods.

**Dialogue state is the bottleneck of LLMs.** As shown in 7 and 8, LLMs do not outperform supervised methods in DST and End-to-end Modeling. But, it is worth noting that LLMs achieve comparable performances in Policy Optimization task. It shows LLMs can complete the user request if provided the golden dialogue state and database results. However, when LLMs use their predicted dialogue state and database results in End-to-end Modeling, the performance is much lower. This suggests that LLM's ability to accomplish dialogue goals depends on the predictions of the dialogue state. To explore the reasons for the poor performance of LLMs, we provide an analysis in Appendix A.9. We find the main reason in DST is that the hallucination phenomenon [23] is very serious, i.e., LLMs generate erroneous results at slots that are not involved in the conversation. Meanwhile, LLMs are sensitive to noisy utterances, e.g., LLMs tend to directly copy the noisy word in the utterance as result, which may be due to the inability of LLMs to perceive the speech information.

# 7 Conclusion

In this paper, we introduce SpokenWOZ, a large-scale spoken Task-oriented Dialogue (TOD) dataset that incorporates both text and speech inputs. SpokenWOZ encompasses with unique characteristics of spoken conversations, including word-by-word processing, ASR noise, and reasoning. In addition, we introduce two new slot types, cross-turn and reasoning slots, as novel challenges. We further present a range of baseline results to demonstrate the usability of the dataset. We hope that SpokenWOZ, with its rich, dual-modal data and considerable volume, will drive the progress of spoken TOD modeling.

## Acknowledgement

This work was supported by Alibaba Group. Meanwhile, we would like to thank Prof. Milica Gasic for her appreciation and advice on our idea at the beginning of the project. We thank Dr. Bowen Yu for his constructive comments on our writing. Finally, we would like to thank all the annotators for their efforts. We look forward to our dataset advancing the research on spoken TOD.

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

# A  Appendix

## A.1  Construction & Schema Details

### A.1.1  Conversation Details

To make spoken conversations that close to the real scenarios, we change the following interaction pattern in MultiWOZ. In SpokenWOZ, once the user's booking is successful, the agent will provide the entity booked and ask for the user's profile information, rather than providing a reference code in MultiWOZ. Profile information including name, ID, email, license plate number, and phone. We will explain in detail when agents will actively collect profile information from users.

**Name.**   When a user makes a successful hotel and restaurant reservation, the agent will request the user's name as the reserved information. The user's name is randomly generated by the script[5].

**ID number.**   When a user books a train, the agent will ask for the user's ID number as registration information, which is a randomly generated 16-digit string.

**Email.**   When the user completes the hotel or restaurant reservation, the agent will ask the user if she/he wants to receive the order via email. If the user agrees to receive the order, the agent will request the user's email. The mailbox number consists of the first letter of the user's first name plus the user's last name, plus four randomly generated characters, and randomly choose one of "@gmail.com", "@yahoo.com", "@outlook.com", "@hotmail.com" as the suffix.

Table 9: The 36 slots are tracked in the dialogue state.

| | |
|---|---|
| *attraction* | area / name / type |
| *hospital* | department |
| *hotel* | area / bookday / bookpeople / bookstay / internet / name / parking / pricerange / stars / type |
| *restaurant* | booktime / bookday / bookpeople / area / food / name / pricerange |
| *taxi* | arriveby / departure / leaveat / destination |
| *train* | arriveby / departure / destination / leaveat / bookpeople / day |
| *profile* | license plate number / name / ID / email / phone |

**License plate number.**   When a user reserves a parking space at a hotel, attraction, or restaurant, the agent will request the user's license plate number. The license plate number is a string of 7 random characters, the first two are letters, the middle two are numbers, and the last three are letters.

**Phone number.**   When a user successfully books a taxi, the agent will request the user's phone number to contact the taxi driver, which is a randomly generated 10-digit string. In another case, when users inquire about police station information, the agent will also ask for the user's phone number as a contact number.

### A.1.2  Slot Details

The following 36 slots are tracked in the dialogue state shown in Table 9. We also list the reasoning slot in Table 10. To control the number of cases where the value needs to be reasoned about in the reasoning slots, we require participants to implicitly express the values specified in the task goal. 20% of the reasoning slot values will be automatically marked as requiring implicit expression in the conversation.

### A.1.3  Speaker Origins Details

Considering the different laws of data access of the different countries, we chose Canada, Singapore, China, South Africa to collect the audio data, which will enable us to open source the audio data. We also found that the

Table 10: Reasoning slot in SpokenWOZ. The upper script indicates which domains it belongs to. *: universal, 1: restaurant, 2: hotel, 3: attraction, 4: taxi, 5: train, 6: hospital, 7: police, 8: profile.

| | |
|---|---|
| *Temporal Reasoning* | leaveat[4,5] arriveby[4,5] booktime[1] / day[5] bookday[1,2] |
| *Mathematical Reasoning* | bookpeople[1,2,5] bookstay[2] |
| *Semantic Reasoning* | type[1,2,3] area[1,2,3] internet[2] department[6] parking[2] |

---

[5]names: https://github.com/treyhunner/names

cost of collecting audio data from Canada and Singapore is about three times that of collecting from South Africa. Therefore, within the same budget, we chose to collect more audios from South Africa, and we believe that a larger data set would further prompt the research in the community. The distribution of speaker origins are shown in Table 11.

### A.1.4 Audio Details

Our audio files are two-track. One track represents the voice of the user and the other represents the voice of the agent. Meanwhile, the sample rate of our audio files is 8000Hz. Each dialogue corresponds to an audio file, and each word is recorded in the text annotation corresponding to the word context, start time and end time. To avoid the problem of overlapping utterances, we follow the rules below during the collection: (i) prohibit the agent from using the backchannel to interrupt the user; (ii) when a user uses a backchannel expression, the agent should respond to the backchannel correctly, rather than ignoring it and continuing the previous utterance. Finally, the word error rate of ASR is 6.1%, calculated from the manually modified agent utterances and the agent utterances recognized by ASR tool.

Table 11: The origins diversity of SpokenWOZ. Participants come from four different countries to improve the diversity of spoken conversations.

| Country | Dialogues | Percentage | People | Percentage |
|---|---|---|---|---|
| Canada | 500 | 8.77% | 60 | 24% |
| Singapore | 500 | 8.77% | 40 | 16% |
| China | 2100 | 36.84% | 30 | 12% |
| South Africa | 2600 | 45.61% | 120 | 48% |

### A.1.5 Data Division

SpokenWOZ is divided into 4200/500/1000 dialogues in order by train/dev/test. More details can be found in Table 12. The results of experiments are evaluated by the test set.

Table 12: Statistics of SpokenWOZ.

| Dataset | Train | Dev | Test |
|---|---|---|---|
| Audio Hours | 183 | 22 | 44 |
| Dialogues | 4,200 | 500 | 1000 |
| Turns | 149,126 | 18,384 | 35,564 |
| Tokens | 1,672,984 | 204,644 | 396,933 |
| Avg. Turns | 35.50 | 36.77 | 35.56 |
| Avg. Tokens | 11.21 | 11.13 | 11.16 |

## A.2 Experiment Details

### A.2.1 DST Baselines

**BERT+TripPy**  TripPy makes use of copy mechanisms to fill slots. A slot is filled by one of three copy mechanisms, including: (1) span prediction: values are directly extracted from the user's utterances; (2) inform operations: a value may be copied from the system's inform operations; (3) slot copy: a value may be copied over from a different slot.

**SPACE+TripPy**  We use SPACE to replace the original encoder BERT in TripPy. SPACE is a semi-supervised pre-trained conversation model learning from large-scale dialogue corpora with limited annotations, which can be effectively fine-tuned on different downstream dialogue tasks.

**SPACE+WavLM+TripPy**  To use both speech and text data, we concatenate the embeddings from SPACE and WavLM. Then we use a Transformer encoder as the fusion layer to allow the interaction between the different modalities. Then we use the fused outputs as the representations in the TripPy.

**UBAR**  UBAR is acquired by fine-tuning the GPT-2 on the sequence of the entire dialogue session which is composed of user utterance, dialogue state, database result, system act, and system response of every dialogue turn. During the inference time, it formulates DST as a sequence-to-sequence task. It takes the current user utterance, dialogue history, and the previously predicted dialogue state as input, and gets the dialogue state of the current user utterance.

**SPACE**  SPACE is a semi-supervised pre-trained conversation model learning from large-scale dialogue corpora, which is based on UniLM [8]. Such as UBAR, we use SPACE as pre-trained language model to fine-tuning on the sequence of the entire dialogue session. During the inference, give the history and user utterance, SPACE generates dialogue states by autoregression. SPACE uses the same special token as UBAR to split user utterances, dialogue state, act and system response.

**SPACE+WavLM**  To utilize the generation ability of the SPACE model and the speech-modal information, we concatenate the user utterance embeddings from SPACE and user audio embeddings from WavLM as new user-side inputs. During the inference, the model uses dual-modal inputs to generate the state by autoregression.

**SPACE+WavLM**aligned    Using the annotations in SpokenWOZ, the text of a word can be aligned with its audio segment. To further explore how to make full use of the speech information, we align the token and audio segment of every word in user utterances. Then we add the text embeddings from SPACE and the corresponding embeddings from WavLM as new user-side embeddings. During the inference, the model uses dual-modal input to generate the dialogue state by autoregression.

**ChatGPT:**    ChatGPT (gpt-3.5-turbo) is a conversational LLM that has been trained by reinforcement learning and instruction tuning [24], demonstrating a surprising ability in completing conversations.

**InstructGPT**$_{003}$**:**    InstructGPT$_{003}$ (text-davinci-003) [24] is a 175B LLM trained by reinforcement learning with human feedback and instruction tuning.

### A.2.2    Response Generation Baselines

**UBAR**    UBAR fine-tunes the GPT-2 on the sequence of the entire dialogue, including user utterance, dialogue state, database result, system act, and system response. During the inference, UBAR uses the fine-tuned GPT-2 to generate responses given different inputs based on different task settings.

**GALAXY**    GALAXY is a pre-trained dialogue model that explicitly learns dialogue policy from limited labeled dialogues and large-scale unlabeled dialogue corpora via semi-supervised learning, which is based on UniLM. GALAXY keeps the same input format as UBAR.

**SPACE**    SPACE is a pre-trained dialogue model that benefiting from large-scale dialogue corpora via multi-task learning, including dialog understanding module, dialog policy module and dialog generation module. SPACE is based on UniLM. SAPCE keeps the same input format as UBAR.

**SPACE+WavLM**    We concatenate the user utterance embeddings from SPACE and user audio embeddings from WavLM as new user-side inputs. During the inference, SPACE+WavLM uses dual-modal input to generate dialogue state, act, and final system response by autoregression.

**SPACE+WavLM**aligned    SPACE+WavLM$_{aligned}$ adds the text embeddings from SPACE and the corresponding embeddings from WavLM as new user-side embeddings. During the inference, the model uses dual-modal input to generate dialogue state, act, and final system response.

### A.2.3    Hyperparameters

For text-modal methods, we use the code and hyperparameters provided by their respective papers. For dual-modal methods, we ues the same hyperparameters as text-modal methods. To the fair comparison, we train all the baselines 10 epoch for DST, 25 epoch for response generation and use the final epoch checkpoint to get the results on SpokenWOZ. The results we report are the average of the results using five different seeds. We trained the baselines in NVIDIA A100 and V100.

### A.3    Case Study

We will show the predicted cases to confirm our insights proposed in section Experiments.

**Supervised generative methods are helpful.**    We give the comparison between the generative-method UBAR and extractive-method BERT+TripPy. Extractive methods can not get the value if it does not directly exists in utterance, such as the reasoning slot. Meanwhile, the generative-method can be robust to ASR noise and modify the wrong word in the utterance to the right one.

Table 13: The Case shows that supervised generative methods are helpful.

| |
|---|
| 🖥️: There is a train, the id is called tr8925, do you want to make a booking. |
| 🧑: Yes, please make a booking for me. |
| 🖥️: Okay. How many people? |
| 🧑: Um let me think ah **it's me and I have six friends with me**. |
| (BERT+TripPy: Train-Bookpeople = none) |
| (UBAR: Train-Bookpeople = 7) |

The value 7 of slot Bookpeople can be correctly predicted by UBAR. However, value 7 is not existing which is predicted wrongly by BERT+TripPy.

Table 14: The Case shows that supervised generative methods are helpful.

| |
| --- |
| 😀: Uh, I'm looking for a particular hotel in cambridge. |
| 🤖: Okay. So may I know its name, please? |
| 😀: Um let me check. I think the name is called **lavelle lodge**. |
| (BERT+TripPy: Hotel-Name = lavelle lodge) |
| (UBAR: Hotel-Name = lovell lodge) |

In this case, the correct name of the hotel " lovel lodge" is not predicted correctly by BERT + TripPy, even it extracts the correct span in utterance. We also find that UBAR can get the name correctly, which shows that generative-method can learn the ability to correct errors from ASR.

Table 15: The Case shows that supervised generative methods are helpful.

| |
| --- |
| 😀: My id is **8 8 7 1 6**. |
| (BERT+TripPy: Profile-ID = None) |
| (UBAR: Profile-ID = 88716) |
| 🤖: You can proceed. |
| 😀: **4 6 8 5 9**. |
| (BERT+TripPy: Profile-ID = None) |
| (UBAR: Profile-ID = 8871646859) |
| 🤖: I am still working with. |
| 😀: **6 3 8 1 4 1**. |
| (BERT+TripPy: Profile-ID = None) |
| (UBAR: Profile-ID = 8871646859638141) |

The generative-method can learn the ability to concatenate the value segment from different turns, which can be hardly learned by the extractive-method BERT + TripPy.

**Dual-modal TOD models is what you need.**   We give the case study and comparison between text-modal methods SPACE + TripPy, SPACE and dual-modal methods SPACE + TripPy + WavLM, SPACE + WavLM. Although the main experimental results reflect that speech information improves overall performance, we are more concerned with the performance of the ASR-sensitive slots.

Table 16: The Case shows that dual-modal TOD models is what you need.

| |
| --- |
| 😀: Good afternoon. Yes. Uh, could you please assist me looking for a particular restaurant, please. |
| 🤖: No problem. Do you have a name for a restaurant? |
| 😀: Yes. um, it's called the **bangkok cutting**. |
| 🤖: **Bangkok city**, okay. Just give me a second. I'll look for it on the system. |
| 😀: Okay, not a problem at all. |
| (SPACE+TripPy: Restaurant-Name = bangkok cutting) |
| (SPACE+WavLM+TripPy: Restaurant-Name = bangkok city ) |
| (SPACE: Restaurant-Name = bangkok city) |
| (SPACE+WavLM: Restaurant-Name = bangkok city ) |

In this case, the value of the slot Name is not correctly predicted by SPACE+TripPy. We find that the span prediction copy mechanism is performed in SPACE+TripPy. However, SPACE+WavLM+TripPy performed the copy mechanism and copy the value from Inform Operations to get right value. This indicates that the speech information can be used to further improve performance.

Table 17: The Case shows that dual-modal TOD models is what you need.

👨‍🦱: I want to find a particular hotel to rise. I remember his name, but I can't find the location.
🤖: So may I have the name of the hotel, please.
👨‍🦱: Uh. the name of the hotel is **work worth house**.
🤖: Okay, so I found a hotel called **warkworth house** for you.
👨‍🦱: Uh, yes, yes. That's the hotel. Thank you.
(SPACE+TripPy: Hotel-Name = None)
(SPACE+WavLM+TripPy: Hotel-Name = work worth house)
(SPACE: Hotel-Name = worth house)
(SPACE+WavLM: Hotel-Name = warkworth house )

In this case, SPACE can not predict the value in the right pattern, but dual-modal SPACE+WavLM+TripPy successfully predict it. This shows that speech modality can also help generative methods learn the correct pattern, even in the presence of ASR noise from user utterances.

### A.4  Heatmap of acts

We show the act flow in SpokenWOZ in Figure 6. Given the user dialogue act, we present the frequency of agent act in heat map. As shown in Figure 5 and 6, SpokenWOZ not only contains more types of acts, but also contains more diverse act flow. It is more difficult for the model to predict the right action and give the right response.

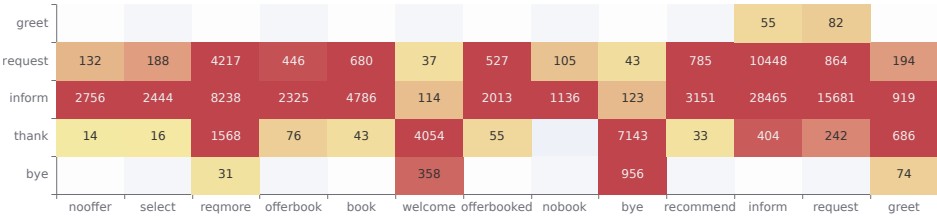

Figure 5: Heat map of agent acts in MultiWOZ. The heat map shows the frequency of the agent act (horizontal axis) after the given user act (vertical axis).

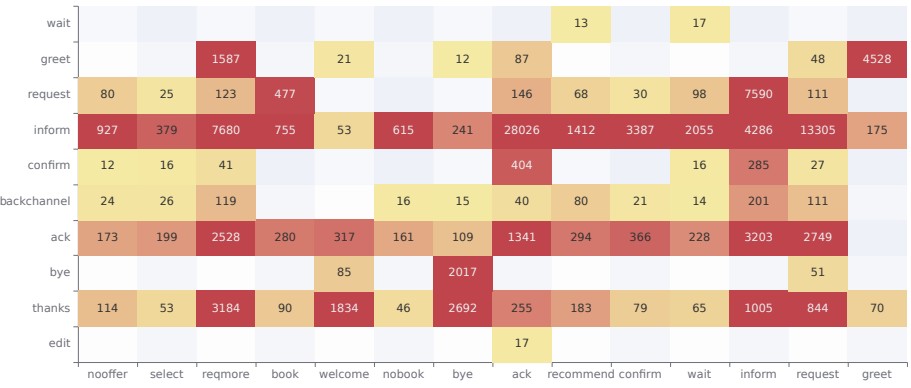

Figure 6: Heat map of agent acts in SpokenWOZ. The heat map shows the frequency of the agent act (horizontal axis) after the given user act (vertical axis).

## A.5  Statistics

We give the distribution of domains in Table 18. Meanwhile, the dataset distributions of dialog length and turn length are shown in the following figures. We give the statistics of SpokenWOZ in Figure 7 and 8. Shown in Figure 7, the length of dialogue history is concentrated above 30 turns. The excessive number of dialogue turns also makes it difficult for the model to learn. We compared the multi-domain and single-domain dialogue in Figure 9, intuitively, the number of turns for multi-domain dialogue is larger than the number of turns for single-domain dialogue. In Figure 10, there is no significant difference between the utterance lengths of the user and agent, because SpokenWOZ is constructed using the Human-to-Human schema. We also show the distribution of the dialogue acts and slots in Figure 11.

Table 18: The distribution of dataset domains.

| Domains | Number |
|---|---|
| profile-restaurant-train | 720 |
| hotel-profile-train | 702 |
| attraction-hotel-profile-taxi | 295 |
| hotel-profile-restaurant-taxi | 294 |
| attraction-profile-train | 291 |
| profile-taxi | 285 |
| attraction-train | 278 |
| attraction-profile-restaurant-taxi | 275 |
| hotel-profile-restaurant | 252 |
| profile-restaurant | 238 |
| attraction | 238 |
| hotel-profile | 237 |
| attraction-hotel-profile | 212 |
| attraction-profile-restaurant | 209 |
| profile-train | 193 |
| train | 149 |
| hotel-train | 148 |
| restaurant-train | 132 |
| hotel | 104 |
| restaurant | 102 |
| attraction-restaurant | 85 |
| attraction-hotel | 62 |
| hospital | 57 |
| police-profile | 56 |
| attraction-profile | 47 |
| hotel-restaurant | 39 |
| Total | 5700 |

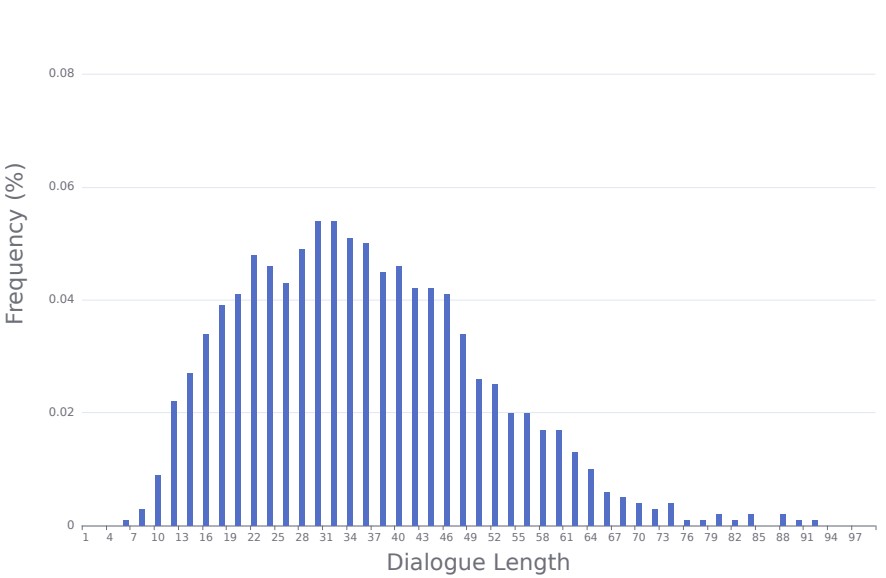

Figure 7: The distribution of the length of turn in SpokenWOZ.

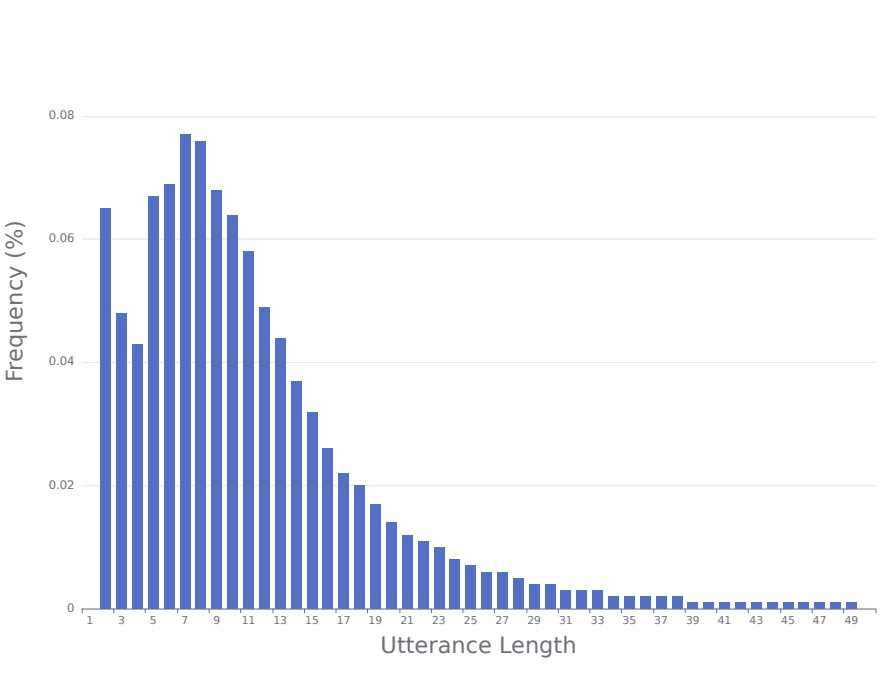

Figure 8: The distribution of the length of turn in SpokenWOZ.

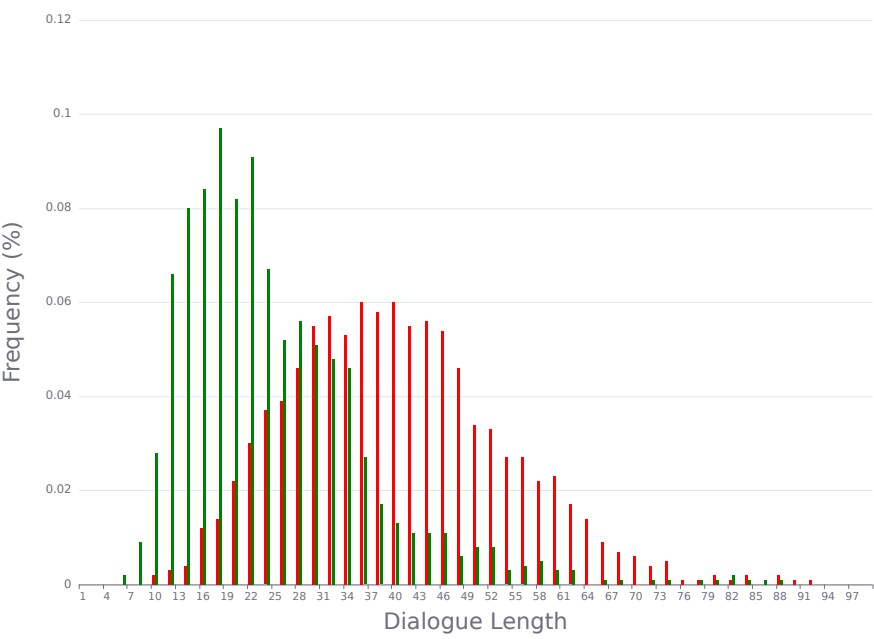

Figure 9: The distribution of the number of turns in two kinds of dialog in SpokenWOZ: Multi-domain, Single-domain.

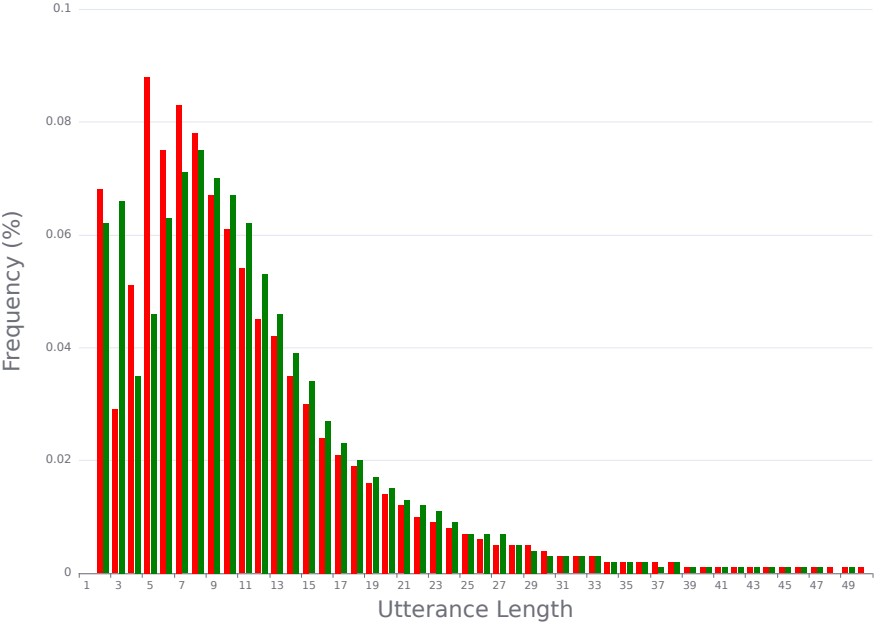

Figure 10: The distribution of the length of turn in two kinds of dialog in SpokenWOZ: User, Agent.

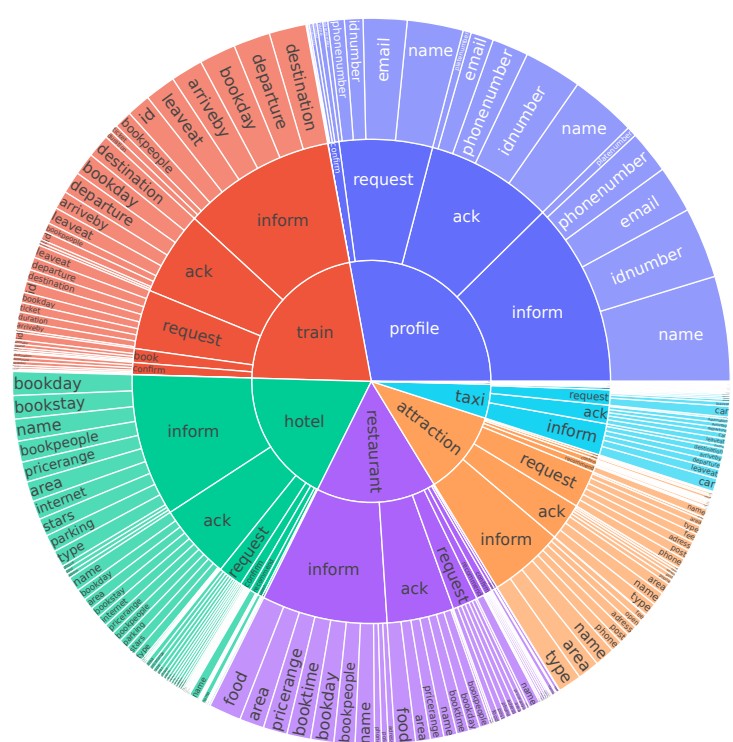

Figure 11: The distribution of domains, dialogue acts and slots in SpokenWOZ.

## A.6 Dialogue Example

A dialogue example can be found in Figure 12.

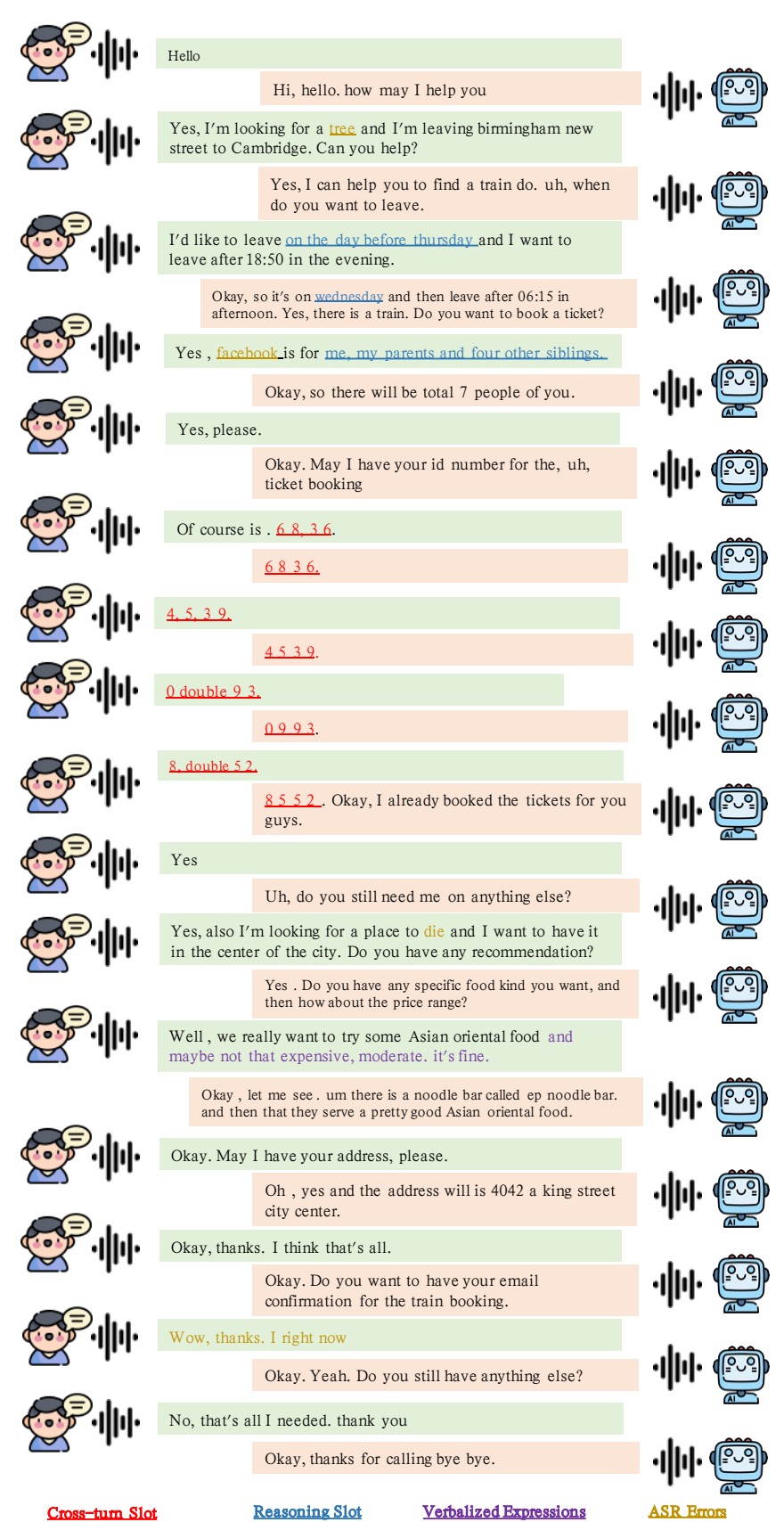

Figure 12: A dialogue example from SpokenWOZ.

## A.7 Online Database

We give the interface of our built database for participants in Figure 13.

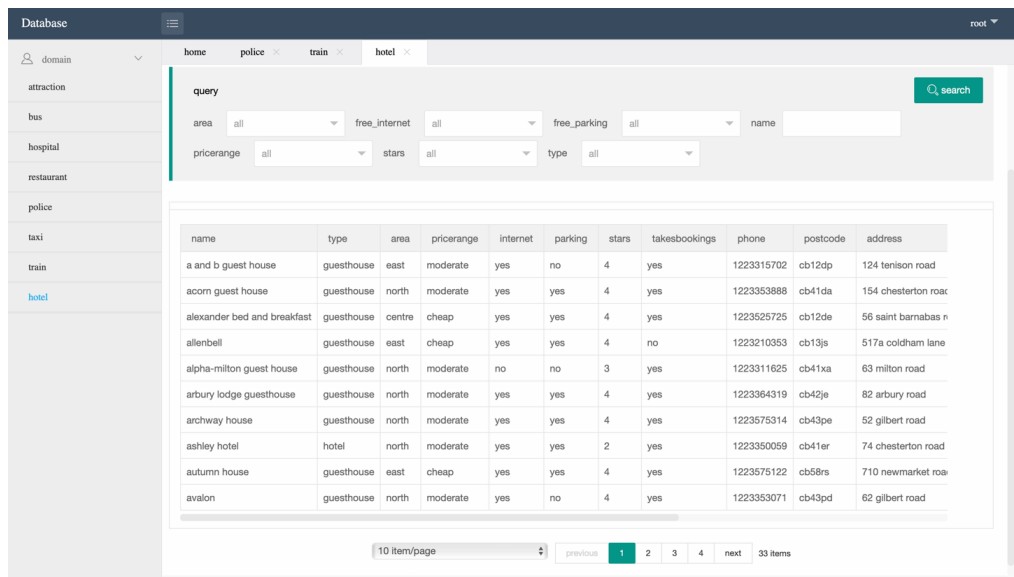

Figure 13: The built online database in SpokenWOZ.

## A.8 Task Goal Example

We give an example of our task goal for participants in Table 19.

Table 19: An example of task goal.

---

(Tips!) The important information is marked with < >. Message with (use veiled expression) means that an veiled expression is needed here.

(Attention!) Please ask customer service for information and try to solve your problem in the Shortest number of turns

(Background) Your name is <Misty Imbert>, your telephone is <9310729130>, your ID number is <8485147021469113>, your email is <MImbert06jn@outlook.com>, your car_number is <OL09ODU>

---

(Background) You are planning your trip in Cambridge

You are looking for a <place to stay>. The hotel <doesn't need to have free parking> and should <include free wifi>

The hotel should be in the <west> and should have <a star of 4>

If there is no such hotel, how about one that has <free parking> Once you find the <hotel> you want to book it for <4 people> and <2 nights> starting from <friday>

You are also looking for a <restaurant>. The restaurant should be in the <moderate> price range and should serve <indian> food

The restaurant should be <in the same area as the hotel>

Once you find the <restaurant> you want to book a table for <the same group of people> at <12:15>(use veiled expression)> on <the same day>

(Background) Once you have made a booking, you do not want to give out your email address for receiving orders.

---

## A.9 Analysis of LLMs

### A.9.1 Prompts for LLMs

We imitate the format of prompt form Hudecek et al. [18] and Bang et al. [1]. We list a prompt example for DST task in Table 20.

Table 20: An example of a zero-shot version of the prompt used for DST.

---

Definition: Give the dialogue state of the last utterance in the following dialogue in JSON (for example: STATE: "hotel-parking": "yes", "hotel-type": "guest house") by using the following pre-defined slots and possible values:

- Slot Name: hotel-pricerange; Slot Descrption: price budget of the hotel; Possible values: ['expensive', 'cheap', 'moderate']
- Slot Name: hotel-type; Slot Descrption: type of the hotel; Possible values: ['guest house', 'hotel']
- Slot Name: hotel-parking; Slot Descrption: whether the hotel has parking; Possible values: ['no', 'yes']
- Slot Name: hotel-day; Slot Descrption: day of the hotel booking; Possible values: ['monday', 'tuesday', 'wednesday', 'thursday', 'friday', 'saturday', 'sunday']
- Slot Name: hotel-people; Slot Descrption: number of people booking the hotel; Possible values: ['1', '2', '3', '4', '5', '6', '7', '8']
- Slot Name: hotel-stay; Slot Descrption: length of stay at the hotel; Possible values: ['1', '2', '3', '4', '5', '6', '7', '8']
- Slot Name: hotel-internet; Slot Descrption: whether the hotel has the free internet; Possible values: ['no', 'yes']
- Slot Name: hotel-name; Slot Descrption: name of the hotel; Possible values: []
- Slot Name: hotel-area; Slot Descrption: area of the hotel; Possible values: ['centre', 'east', 'north', 'south', 'west']
- Slot Name: hotel-star; Slot Descrption: star of the hotel;Possible values: ['0', '1', '2', '3', '4', '5']
- Slot Name: train-arriveby; Slot Descrption: the arrival time of the train, 24-hour standard time, e.g. 06:00, 18:30; Possible values: []
- Slot Name: train-day; Slot Descrption: day of the train departure; Possible values: ['monday', 'tuesday', 'wednesday', 'thursday', 'friday', 'saturday', 'sunday']
- Slot Name: train-people; Slot Descrption: number of people travelling by train; Possible values: ['1', '2', '3', '4', '5', '6', '7', '8']
- Slot Name: train-leaveat; Slot Descrption: leaving time of the train, 24-hour standard time, e.g. 06:00, 18:30; Possible values: []
- Slot Name: train-destination; Slot Descrption: destination of the train; Possible values: ['birmingham new street', 'bishops stortford', 'broxbourne', 'cambridge', 'ely', 'kings lynn', 'leicester', 'london kings cross', 'london liverpool street', 'norwich', 'peterborough', 'stansted airport', 'stevenage']
- Slot Name: train-departure; Slot Descrption: departure of the train; Possible values: ['birmingham new street', 'bishops stortford', 'broxbourne', 'cambridge', 'ely', 'kings lynn', 'leicester', 'london kings cross', 'london liverpool street', 'norwich', 'peterborough', 'stansted airport', 'stevenage']
- Slot Name: attraction-area; Slot Descrption: area of the attraction; Possible values: ['centre', 'east', 'north', 'south', 'west']
- Slot Name: attraction-name; Slot Descrption: name of the attraction; Possible values: []
- Slot Name: attraction-type; Slot Descrption: type of the attraction; Possible values: ['architecture', 'boat', 'cinema', 'college', 'concerthall', 'entertainment', 'museum', 'multiple sports', 'nightclub', 'park', 'swimmingpool', 'theatre']
- Slot Name: restaurant-pricerange; Slot Descrption: price budget for the restaurant; Possible values: ['expensive', 'cheap', 'moderate']
- Slot Name: restaurant-area; Slot Descrption: area of the restaurant; Possible values: ['centre', 'east', 'north', 'south', 'west']
- Slot Name: restaurant-food; Slot Descrption: the cuisine of the restaurant; Possible values: []
- Slot Name: restaurant-name; Slot Descrption: name of the restaurant; Possible values: []
- Slot Name: restaurant-day; Slot Descrption: day of the restaurant booking; Possible values: ['monday', 'tuesday', 'wednesday', 'thursday', 'friday', 'saturday', 'sunday']
- Slot Name: restaurant-people; Slot Descrption: number of people for the restaurant booking; Possible values: ['1', '2', '3', '4', '5', '6', '7', '8']
- Slot Name: restaurant-time; Slot Descrption: time of the restaurant booking, 24-hour standard time, e.g. 06:00, 18:30; Possible values: []
- Slot Name: hospital-department; Slot Descrption: department of the hospital; Possible values: []
- Slot Name: taxi-leaveat; Slot Descrption: leaving time of taxi, 24-hour standard time, e.g. 06:00, 18:30; Possible values: []
- Slot Name: taxi-destination; Slot Descrption: destination of taxi; Possible values: []
- Slot Name: taxi-departure; Slot Descrption: departure location of taxi; Possible values: []
- Slot Name: taxi-arriveby; Slot Descrption: arrival time of taxi, 24-hour standard time, e.g. 06:00, 18:30; Possible values: []
- Slot Name: profile-name; Slot Descrption: the name of the user; Possible values: []
- Slot Name: profile-email; Slot Descrption: the email of the user; Possible values: []
- Slot Name: profile-idnumber; Slot Descrption: the idnumber of the user; Possible values: []
- Slot Name: profile-phonenumber; Slot Descrption: the phonenumber of the user; Possible values: []
- Slot Name: profile-platenumber; Slot Descrption: the platenumber of the user; Possible values: []

USER: Hello, I need some help.
SYSTEM: Okay, how can i help you.
USER: Yes, I'm looking for a train.

STATE:

---

For response generation tasks, we follow the prompt as Hudecek et al. [18]. We use the dialogue state to query the database provided to get the entity. For the Policy Optimization task, we use the ground truth dialogue state to query the provided database. For End-to-end Modeling, we use the predicted dialogue state generated from ChatGPT or InstructGPT$_{003}$ to query the provided database. We use the contents of the database and the

generated system response to match the strings and complete the de-lexicalization process. The de-lexicalized response will be used to calculate INFORM and SUCCESS. We list a prompt example for a response generation task in Table 21.

Table 21: An example of a zero-shot version of the prompt used for Response Generation.

Definition: Please continue the dialogue as a task-oriented dialogue system called SYSTEM. The answer of SYSTEM should follow the DATABASE provided in JSON format and answer the USER's last utterance. SYSTEM can recommend and inform the contents in the DATABASE according to the utterance of the USER and return the name of the entity when it comes to restaurants, hotels and attractions, and the trainid when it comes to trains. But only when the USER requests information about an entity in the DATABASE, such as a phone number, should SYSTEM inform the corresponding content.

DATABASE:
{ "restaurant": { "address": "Cambridge Retail Park Newmarket Road Fen Ditton", "area": "east", "food": "italian", "name": "pizza hut fen ditton", "phone": "12233237370", "postcode": "cb58wr", "pricerange": "moderate" } }

DIALOGUE CONTENT:
USER: Hello.
SYSTEM: Hello! How can I assist you today? Are you looking for any specific information or recommendations?
USER: I'm in east part of the city and I want to have some food. Uh, hope for the. to serve Italian food and with moderate price range.

SYSTEM:

## A.9.2 Analysis on DST

As shown in the section Experiments, the performances of LLMs do not surpass that of supervised small models and show a noticeable gap compared to the supervised generative dual-modal baselines in the DST task. However, it is worth noting that LLMs slightly outperform the BERT+TripPy when cross-turn slots are not taken into account, showing the potential capabilities of LLMs. Meanwhile, we find that the main reason for the poor results of LLMs is that the hallucination phenomenon [23] is very serious, e.g., LLMs often generate additional results that do not fit the dialogue utterances. When we measure only the slots where the ground truth value is not "none", we find that ChatGPT achieves JGA of 30.81 and InstructGPT$_{003}$ achieves JGA of 34.42. Compared to the standard JGA, ChatGPT and InstructGPT$_{003}$ improve their scores by 17.06 and 20.27 respectively. This indicates that LLMs generate a large number of erroneous results at slots that are not involved in the conversation, i.e., the hallucination phenomenon is very serious. We show a case in Table 24.

Table 22: The Case shows the hallucination phenomenon.

: Hello.
: Hello, how can I help?
: Yes, I'm looking for restaurant.
(ChatGPT: Restaurant-Food = international)
(Ground Truth: Restaurant-Food = none)

Meanwhile, due to the inability of LLMs to perceive the information of speech, LLMs tend to generate the value directly from user utterance. We show a case in Table 23.

Table 23: The Case shows that LLMs are sensitive to the noisy utterance.

: Uh, I'm looking for a particular hotel in cambridge.
: Okay. So may I know its name, please?
: Um let me check. I think the name is called **lavelle lodge**.
(ChatGPT: Hotel-Name = lavelle lodge)
(InstructGPT$_{003}$: Hotel-Name = lavelle lodge)
(Ground Truth: Hotel-Name = lovell lodge)

## A.9.3 Analysis on Response Generation

As introduced in the section Experiments, LLMs achieve comparable performances in Policy Optimization task but poor performances in the End-to-end Modeling task. We find that the main reason for the poor results in End-to-end Modeling is that the entity returned by the database does not meet the user's needs. Meanwhile,

the poor performances of BLEU show that there is a big difference between LLM's response style and human response style. Meanwhile, LLMs may feel confused about noisy utterances and generate a statement requesting clarification as shown in Table 24.

Table 24: The Case shows that LLMs is sensitive to the noisy utterance.

👤: Mm. I'm looking for a place to die (*In the audio it is actually "dinner"*).
ChatGPT: I'm sorry, I'm not sure I understand. Could you please rephrase your question?

# B    Limitations

Even though we tried hard to build a realistic spoken TOD benchmark for further studies, we could not use the audio data from real conversations due to privacy concerns. This brings the following limitations: (1) For "profile" domain, we use a designed script to generate random personal information, which may not be realistic, and the number of slots also limits the further in scenario that agents need to collect personal information from users; (2) As the limited ontology, our benchmark should mainly be used for research instead of deployed in realistic applications.

# C    Ethics Statement

We construct SpokenWOZ, a task-oriented dialogue benchmark containing both audio data and text data, and we will detail our ethical considerations for each part of our collection process:

**Ontology Consideration.**    We inherited and expanded MultiWOZ's ontology, which is open-source and under the MIT License. We have used it in compliance with its terms of use.

**Privacy Concern.**    In our dataset, we have designed the scenarios where an agent needs to proactively collects user information, however, our user information is all generated by scripts in a random manner, so there will not be any privacy leakage issues.

**Audio Collection.**    We informed each participant that the collected audio data will be used as public dataset for research. Participants who agree to participate in data collection will sign a contract with us, and the ownership and use rights of their data belong to us. We will not disclose which specific participant the audio came from. At the same time, we have reviewed the legal regulations of four countries and regions, and will release the data in a legal manner, so this will not cause any legal problems. The distribution of our data sources has been discussed in the Appendix A.1.3, the diversity of SpokenWOZ makes our data unbiased. We pay $30k for 249 hours audio. The average cost per hour of audio is $120.

**Dialogue Annotation.**    During the annotation process, the annotators' personal information is not collected, which will not cause privacy leakage. At the same time, during the annotation process, we signed a non-disclosure agreement with each annotator, therefore, the audio data will not be leaked during the annotation process. We pay $20k for 5,700 dialogues. The average cost per dialogue annotation is $3.5. After our statistics, an average of one hour can annotate 5 dialogues.

# D    Data Format

## D.1    Audio Format

As detailed in Appendix A.1.4, audio files are two-track with a sample rate of 8000Hz. One track represents the voice of the user and the other represents the voice of the agent. Each dialogue corresponds to an audio file, and each word is recorded in the text annotation corresponding to the word context, start time, and end time. We use the wav format to save our audio files. The file name of the audio is consistent with the id of the dialogue, for example, the corresponding audio file for MUL0032 is MUL0032.wav.

## D.2    Text Format

Our text data is given in json format, and we take the same fields as popular MultiWOZ 2.2 [42] to store the corresponding information, so researchers can easily use our data. In addition, we additionally provide the data ontology and the database json files. There are 5,700 dialogues ranging form single-domain to multi-domain in SpokenWOZ. The test sets contain 1k examples. Dialogues with MUL in the name refers to multi-domain dialogues. Dialogues with SNG refers to single-domain dialogues. Each dialogue consists of a goal, multiple user and system utterances, dialogue state, dialogue act, corresponding audio and ASR transcription.

The dialogue goal for each dialogue is recorded in the "goal" field. The dialogue goal holds the fields involved in the dialogue as well as the slots involved and the corresponding values.

The dialogue state for each dialogue is recorded in the "metadata" field in every turn the same as MultiWOZ 2.2. The state have two sections: semi, book. Semi refers to slots from a particular domain. Book refers to booking slots for a particular domain. The joint accuracy metrics includes ALL slots.

The dialogue act for each dialogue is recorded in the "dialogue_act" and "span_info" field in every turn:

```
{
  "$dialogue_id": {
  "log":{
    "$turn_id": {
      "dialogue_act": {
        "$act_name": [
          [
            "$slot_name",
            "$action_value"
          ]
        ]
      },
      "span_info": [
        [
          "$act_name"
          "$slot_name",
          "$action_value"
          "$start_charater_index",
          "$exclusive_end_character_index"
        ]
  }
}
```

The ASR transcription for each dialogue is recorded in the "words" field in every turn.

```
{
  "$dialogue_id": {
  "log":{
    "$turn_id": {
      "words": [
        {
        "$word_context": "$word",
        "$begin_time": "$begintime",
        "end_time": "$endtime",
        "channel_id": "$channel",
        "word_index": "$index",
        }
  }
}
```

# E    Datasheets for SpokenWOZ

## E.1    Dataset documentation and intended uses

**For what purpose was the dataset created? Was there a specific task in mind? Was there a specific gap that needed to be filled? Please provide a description.**    Task-oriented dialogue (TOD) models have made significant progress in recent years. These systems are designed to assist users in accomplishing specific goals, e.g., flight booking and restaurant reservation. However, these TOD datasets constructed solely based on written texts may not accurately reflect the nuances of spoken conversations, leading to a gap between academic research and real-world spoken TOD scenarios. We introduce the common tasks of TOD, including dialogue state tracking, policy Optimization, and End-to-end Modeling.

**Who created this dataset (e.g., which team, research group) and on behalf of which entity (e.g., 15 company, institution, organization)?**    This dataset is created by researchers at Alibaba Group, Renmin University of China, and University of Michigan.

**Who funded the creation of the dataset?**  The creation of dataset was funded by DAMO Academy, Alibaba Group.

**Any other comments?**  N/A

### E.2  Composition

**What do the instances that comprise the dataset represent (e.g., documents, photos, people, countries)? Are there multiple types of instances (e.g., movies, users, and ratings; people and interactions between them; nodes and edges)? Please provide a description.**  The dataset contains text data and aduio data of spoken task-oriented dialogue. For each dialogue text, we also annotate both the dialogue state and dialogue act. For the dialogue audio, we also give the ASR transcription and audio file.

**How many instances are there in total (of each type, if appropriate)?**  SpokenWOZ contains 8 domains, 203k turns, 5.7k dialogues and 249 hours of audios from spoken conversations.

**What data does each instance consist of? "Raw" data (e.g., unprocessed text or images) or features? In either case, please provide a description.**  For a conversation data, it includes a text part and a audio part. For each dialogue text, SpokenWOZ contains dialogue context, annotated dialogue state and annotated dialogue act. For the dialogue audio, SpokenWOZ contains corresponding audio data and ASR transcription.

**Is there a label or target associated with each instance?If so, please provide a description**  Yes, we inherit and extend the MultiWOZ annotation schema, which is widely used for task-oriented dialogue.

**Is any information missing from individual instances? If so, please provide a description, explaining why this information is missing (e.g., because it was unavailable). This does not include intentionally removed information, but might include, e.g., redacted text.**  For individual instances, there is no missing information.

**Are relationships between individual instances made explicit (e.g., users' movie ratings, social network links)? If so, please describe how these relationships are made explicit.**  Each dialogue in SpokenWOZ is relatively independent, and the domains involved are different.

**Are there recommended data splits (e.g., training, development/validation,testing)? If so, please provide a description of these splits, explaining the rationale behind them.**  We give the data splits in Appendix A.1.5. The data is split into training, development, and unreleased test sets. Once researchers have built a model that works to your expectations on the dev set, they can submit it to us to get official scores on the hidden test set. To mitigate the misestimation of the generalization error of the model, we increase the number of test set to 1000 dialogues. At the same time we keep the training and test data domain distributions roughly, but not exactly, the same.

**Are there any errors, sources of noise, or redundancies in the dataset? If so, please provide a description.**  Since our dataset requires manual annotation of dialogue state and dialogue act, annotation noise is inevitably introduced. At the same time the dialogue audio collection there are cases of substandard audio quality, such as low communication quality. As shown in section SpokenWOZ Construction, strict quality control is performed at each collection stage.

**Is the dataset self-contained, or does it link to or otherwise rely on external resources (e.g., websites, tweets, other datasets)?**  SpokenWOZ is self-contained.

**Does the dataset contain data that might be considered confidential (e.g., data that is protected by legal privilege or by doctor-patient confidentiality, data that includes the content of individuals' non-public communications)?If so, please provide a description.**  No.

**Does the dataset relate to people? If not, you may skip the remaining questions in this section.**
No. Detailed in Ethics Statement, we have designed the scenarios where an agent needs to proactively collects user information, however, our user information is all generated by scripts in a andom manner, so there will not be any privacy leakage issues.

### E.3 Collection process

**How was the data associated with each instance acquired? Was the data directly observable (e.g., raw text, movie ratings), reported by subjects (e.g., survey responses), or indirectly inferred/derived from other data (e.g., part-of-speech tags, model-based guesses for age or language)? If data was reported by subjects or indirectly inferred/derived from other data, was the data validated/verified? If so, please describe how.** We report the construction schema of SpokenWOZ in the section SpokenWOZ Construction.

**What mechanisms or procedures were used to collect the data (e.g., hardware apparatus or sensor, manual human curation, software program, software API)? How were these mechanisms or procedures validated?** We organized 250 participants to generate 5,700 dialogues via phone calls. The details of the audio file can be found in Appendix A.1.4. The dialogue state and dialogue act are annotated using the Appen platform[6].

**If the dataset is a sample from a larger set, what was the sampling strategy (e.g., deterministic, probabilistic with specific sampling probabilities)?** SpokenWOZ is not sampled from a larger set.

**Over what timeframe was the data collected? Does this timeframe match the creation timeframe of the data associated with the instances (e.g., recent crawl of old news articles)? If not, please describe the time-frame in which the data associated with the instances was created.** Our data collection starts in July 2022. The contents of our data instances are independent of the time of collection.

**Were any ethical review processes conducted (e.g., by an institutional review board)? If so, please provide a description of these review processes, including the outcomes, as well as a link or other access point to any supporting documentation.** Not applicable. We consider these contents in Ethics Statement.

**Does the dataset relate to people? If not, you may skip the remainder of the questions in this section.** No.

### E.4 Preprocessing/cleaning/labeling

**Was any preprocessing/cleaning/labeling of the data done (e.g., discretization or bucketing, tokenization, part-of-speech tagging, SIFT feature extraction, removal of instances, processing of missing values)? If so, please provide a description. If not, you may skip the remainder of the questions in this section.** We report the construction schema of SpokenWOZ in the section SpokenWOZ Construction.

**Was the "raw" data saved in addition to the preprocessed/cleaned/labeled data (e.g., to support unanticipated future uses)? If so, please provide a link or other access point to the "raw" data.** Raw data was not saved to prevent misuse. We will only open source the cleaned data.

**Is the software used to preprocess/clean/label the instances available? If so, please provide a link or other access point.** We have used Python language to implement data cleaning. We will share the scripts details in our codebase.

### E.5 Uses

**Has the dataset been used for any tasks already? If so, please provide a description.** The complexity and diverse spoken characteristics in SpokenWOZ make it a useful dataset for different TOD tasks, including dialogue state tracking and response generation. For response generation, the challenges are twofold: Policy Optimization and End-to-end Modeling. More details can be found in section Tasks & Settings.

**Is there a repository that links to any or all papers or systems that use the dataset? If so, please provide a link or other access point.** We provide links to the papers of all the baseline models on the leaderboard[7] we built.

**What (other) tasks could the dataset be used for?** The dataset can be used for the full range of tasks related to task-oriented dialogue and can be used for dual-modal task-oriented dialogue studies.

---

[6]https://appen.com/
[7]https://spokenwoz.github.io/SpokenWOZ-github.io/

**Is there anything about the composition of the dataset or the way it was collected and prepro-cessed/cleaned/labeled that might impact future uses? For example, is there anything that a future user might need to know to avoid uses that could result in unfair treatment of individuals or groups (e.g., stereotyping, quality of service issues) or other undesirable harms (e.g., financial harms, legal risks) If so, please provide a description. Is there anything a future user could do to mitigate these undesirable harms?** NA.

**Are there tasks for which the dataset should not be used? If so, please provide a description.** NA.

## E.6 Distribution

**Will the dataset be distributed to third parties outside of the entity (e.g., company, institution, organization) on behalf of which the dataset was created?If so, please provide a description.** SpokenWOZ dataset and codebases for reproducing the experiments are available at: https://spokenwoz.github.io/SpokenWOZ-github.io/.

**How will the dataset will be distributed (e.g., tarball on website, API, GitHub)? Does the dataset have a digital object identifier (DOI)?** The dataset is now available at: https://spokenwoz.github.io/SpokenWOZ-github.io/.

**When will the dataset be distributed?** The dataset is available now.

**Will the dataset be distributed under a copyright or other intellectual property (IP) license, and/or under applicable terms of use (ToU)? If so, please describe this license and/or ToU, and provide a link or other access point to, or otherwise reproduce, any relevant licensing terms or ToU, as well as any fees associated with these restrictions.** SpokenWOZ is distributed under the CC BY-NC 4.0 [8] license. CC BY-NC 4.0 allows reusers to distribute, remix, adapt, and build upon the material in any medium or format for noncommercial purposes only, and only so long as attribution is given to the creator. If you remix, adapt, or build upon the material, you must license the modified material under identical terms.

**Have any third parties imposed IP-based or other restrictions on the data associated with the instances? If so, please describe these restrictions, and provide a link or other access point to, or otherwise reproduce, any relevant licensing terms, as well as any fees associated with these restrictions.** No.

**Do any export controls or other regulatory restrictions apply to the dataset or to individual instances? If so, please describe these restrictions, and provide a link or other access point to, or otherwise reproduce, any supporting documentation.** No.

## E.7 Maintenance

**Who is supporting/hosting/maintaining the dataset?** Authors of this work bear all responsibility in case of violation of rights. Shuzheng Si (sishuzheng@foxmail.com) and Wentao Ma (mawentao.mwt@alibaba-inc.com) will be responsible for maintaining this dataset.

**How can the owner/curator/manager of the dataset be contacted (e.g., email address)?** If you wish to extend or contribute to our dataset SpokenWOZ, please contact us via email - Shuzheng Si (sishuzheng@foxmail.com) and Wentao Ma (mawentao.mwt@alibaba-inc.com).

**Is there an erratum? If so, please provide a link or other access point.** Any updates to the dataset wiil be shared via GitHub

**Will the dataset be updated (e.g., to correct labeling errors, add new instances, delete instances)? If so, please describe how often, by whom, and how updates will be communicated to users (e.g.,mailing list,GitHub)?** If we find inconsistencies in the dataset or extend the dataset, we will release the new version on the website and Github.

**If the dataset relates to people, are there applicable limits on the retention of the data associated with the instances (e.g., were individuals in question told that their data would be retained for a fixed period of time and then deleted)?** N/A

---

[8] https://creativecommons.org/licenses/by-nc/4.0/legalcode

**Will older versions of the dataset continue to be supported/hosted/maintained? If so, please describe how. If not, please describe how its obsolescence will be communicated to users.** All versions of SpokenWOZ will be continue to be supported and maintained on website. We will post the updates on the website and Github.

**If others want to extend/augment/build on/contribute to the dataset, is there a mechanism for them to do so? If so, please provide a description. Will these contributions be validated/verified? If so, please describe how. If not, why not? Is there a process for communicating/distributing these contributions to other users? If so, please provide a description.** Yes. Please contact the authors of this paper for building upon this dataset.

### E.8 Responsibility

The authors bear all responsibility in case of violation of rights, etc. We confirm that the dataset is licensed under CC BY-NC 4.0 license.

