# Supplementary Materials

## A    Appendix

### A.1    Construction & Schema Details

#### A.1.1    Conversation Details

To make spoken conversations that close to the real scenarios, we change the following interaction pattern in MultiWOZ. In SpokenWOZ, once the user's booking is successful, the agent will provide the entity booked and ask for the user's profile information, rather than providing a reference code in MultiWOZ. Profile information including name, ID, email, license plate number, and phone. We will explain in detail when agents will actively collect profile information from users.

**Name.**    When a user makes a successful hotel and restaurant reservation, the agent will request the user's name as the reserved information. The user's name is randomly generated by the script[1].

**ID number.**    When a user books a train, the agent will ask for the user's ID number as registration information, which is a randomly generated 16-digit string.

**Email.**    When the user completes the hotel or restaurant reservation, the agent will ask the user if she/he wants to receive the order via email.  If the user agrees to receive the order, the agent will request the user's email.  The mailbox number consists of the first letter of the user's first name plus the user's last name, plus four randomly generated characters, and randomly choose one of "@gmail.com", "@yahoo.com", "@outlook.com", "@hotmail.com" as the suffix.

Table 1: The 36 slots are tracked in the dialogue state.

| | |
|---|---|
| *attraction* | area / name / type |
| *hospital* | department |
| *hotel* | area / bookday / bookpeople / bookstay / internet / name / parking / pricerange / stars / type |
| *restaurant* | booktime / bookday / bookpeople / area / food / name / pricerange |
| *taxi* | arriveby / departure / leaveat / destination |
| *train* | arriveby / departure / destination / leaveat / bookpeople / day |
| *profile* | license plate number / name / ID / email / phone |

**License plate number.**    When a user reserves a parking space at a hotel, attraction, or restaurant, the agent will request the user's license plate number. The license plate number is a string of 7 random characters, the first two are letters, the middle two are numbers, and the last three are letters.

**Phone number.**    When a user successfully books a taxi, the agent will request the user's phone number to contact the taxi driver, which is a randomly generated 10-digit string. In another case, when users inquire about police station information, the agent will also ask for the user's phone number as a contact number.

---

[1]names: https://github.com/treyhunner/names

Submitted to the 37th Conference on Neural Information Processing Systems (NeurIPS 2023) Track on Datasets and Benchmarks. Do not distribute.

### A.1.2 Slot Details

The following 36 slots are tracked in the dialogue state shown in Table 1. We also list the reasoning slot in Table 2. To control the number of cases where the value needs to be reasoned about in the reasoning slots, we require participants to implicitly express the values specified in the task goal. 20% of the reasoning slot values will be automatically marked as requiring implicit expression in the conversation. Meanwhile, co-reference annotation is already present in SpokenWOZ. Instead of annotating pronouns, we directly annotate the appropriate value in the corresponding slot.

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

**Hello**

> Hi, hello. how may I help you

Yes, I'm looking for a _tree_ and I'm leaving birmingham new street to Cambridge. Can you help?

> Yes, I can help you to find a train do. uh, when do you want to leave.

I'd like to leave on the day before thursday and I want to leave after 18:50 in the evening.

> Okay, so it's on wednesday and then leave after 06:15 in afternoon. Yes, there is a train. Do you want to book a ticket?

Yes , _facebook_ is for me, my parents and four other siblings.

> Okay, so there will be total 7 people of you.

Yes, please.

> Okay. May I have your id number for the, uh, ticket booking

Of course is . 6 8, 3 6.

> 6 8 3 6.

4, 5, 3 9.

> 4 5 3 9.

0 double 9 3.

> 0 9 9 3.

8, double 5 2.

> 8 5 5 2 . Okay, I already booked the tickets for you guys.

Yes

> Uh, do you still need me on anything else?

Yes, also I'm looking for a place to _die_ and I want to have it in the center of the city. Do you have any recommendation?

> Yes . Do you have any specific food kind you want, and then how about the price range?

Well , we really want to try some Asian oriental food and maybe not that expensive, moderate. it's fine.

> Okay , let me see . um there is a noodle bar called ep noodle bar. and then that they serve a pretty good Asian oriental food.

Okay. May I have your address, please.

> Oh , yes and the address will is 4042 a king street city center.

Okay, thanks. I think that's all.

> Okay. Do you want to have your email confirmation for the train booking.

Wow, thanks. I right now

> Okay. Yeah. Do you still have anything else?

No, that's all I needed. thank you

> Okay, thanks for calling bye bye.

**Cross–turn Slot**    **Reasoning Slot**    **Verbalized Expressions**    **ASR Errors**