# OpenReview forum: "SpokenWOZ: A Large-Scale Speech-Text Benchmark for Spoken Task-Oriented Dialogue Agents"
_NeurIPS.cc/2023/Track/Datasets_and_Benchmarks — NeurIPS 2023 Datasets and Benchmarks Poster_

### Official Review · Reviewer_bThz · 2023-06-30
**Very large valuable spoken dialogue dataset**

**Rating:** 6
**Confidence:** 3
**Correctness:** Yes
**Clarity:** No. Please see "Opportunities For Imp…

**Strengths:**

1. The dataset can be used for advancing spoken TOD research and its related area.
2. The evaluation results revealed current LLMs' ability of TOD understanding.

**Additional Feedback:**

As a speech researcher, I'd like to see the prosodic variations of the collected dialogue recordings aggregated by the dialogue act tags.

**Documentation:**

Yes

**Limitations:**

Yes

**Opportunities For Improvement:**

1. The details of qualification test written in Section 3.1 is unclear. Please explain the reason why the authors had to filter out only 250 participants from total 1520 applications.
2. The explanation of quality control described in Section 3.1 is also unclear. What kind of criteria did the authors use to judge that the recorded dialogue was "poor communication quality?"
3. The speaker origins details are described in Appendix A.1.3, but other speaker attributes, such as the speakers' gender and age, are not given.
4. In Section 6.1, please explain the reason why the authors selected the models (from BERT+TripPy to InstructGPT$_{003}$) for the comparison.

**Relation To Prior Work:**

Yes

**Summary And Contributions:**

This paper presented SpokenWOZ, a new large-scale spoken task-oriented dialogue dataset including total 249 hours of audios from human-to-human conversations. The main contributions are summarized as follows:

1. Constructing a large-scale spoken TOD dataset containing more than 203K annotated utterances, 5,700 dialogues, and the associated 249 hours of audi
2. Identifying the new challenges in spoken conversation based on a comprehensive analysis of the proposed SpokenWOZ, including cross-turn slot detection and reasoning slot detection
3. Conducting comprehensive experiments on various baselines, including text-modal TOD baselines, newly proposed cross-modal models, and LLM’s  like ChatGPT

---

> ### Author Response · Authors · 2023-08-21
>
> Thanks for your constructive reviews. It is encouraging to see you find our work would be used for advancing spoken task-oriented dialogue research and its related area.
>
> Regarding your concerns:
>
> 1. In fact, we receive 1,520 applications, we conduct a qualification test for those applicants, and only 250 participants pass the test. We do not limit the number of participants to 250. Criteria for the qualification test include whether the task goal is completed and whether the dialogue audio is natural. They are used to test whether the participants understand the process and rules we used to collect the dialogue audio and whether they could ensure the high quality of the dialogue audio that are subsequently collected.
>
> 2. We employ crowd-sourcing to evaluate the quality of each audio and remove the audio that exhibited poor communication quality or did not fulfill the task goal. Specifically, poor communication quality is defined as the presence of at least one inaudible utterance in that audio due to signal problems.
>
> 3. We do not collect private personal information in consideration of the laws and regulations of each region. Meanwhile, we assume that gender and age may not be directly related to audio quality.
>
> 4. In Section 6.1, we choose models including different sizes from 0.1B to 175B, for the LLMs above 10B, we test its zero-shot capability. For the fine-tuned small models, we mainly consider the following:
>
>      a. Comparisons between extractive and generative models.  Specifically, the extractive models include BERT+TripPy, SPACE+TripPy, SPACE+WavLM+TripPy. The generative models include UBAR, SPACE, SPACE+WavLM, SPACE+WavLM_{align}.
>
>      b. Comparisons between text-modal and dual-modal models, including comparisons between text-modal SPACE+TripPy and dual-modal SPACE+WavLM+TripPy, as well as between text-modal SPACE and dual-modal SPACE+WavLM, dual-modal SPACE+WavLM_{align}.
>
>      c. Comparison between different pre-trained models: including comparison between SPACE+TripPy and BERT+TripPy, and comparison between SPACE and UBAR.
>
> 5. We provide detailed statistics about audio-related features on the SpokenWOZ dataset grouped by dialogue acts, including pitch, intensity(RMSE), speech rate, and duration(in seconds).
>
>
> |  Dialog Act | Pitch | Intensity | Speech Rate | Duration(s) |
> |:-----------:|:-----:|:---------:|:-----------:|:-----------:|
> |    greet    |  21.5 |   0.034   |     6.0     |     2.4     |
> |   reqmore   |  19.3 |   0.031   |     6.3     |     3.5     |
> |    inform   |  23.3 |   0.036   |     5.7     |     5.2     |
> |   request   |  20.9 |   0.035   |     6.1     |     4.1     |
> |    thanks   |  21.7 |   0.032   |     6.1     |     2.7     |
> |     bye     |  17.4 |   0.030   |     5.8     |     2.4     |
> |     ack     |  21.0 |   0.030   |     5.9     |     3.9     |
> |     wait    |  19.7 |   0.028   |     6.1     |     4.8     |
> |    select   |  22.1 |   0.028   |     5.8     |     7.8     |
> |   confirm   |  21.5 |   0.032   |     5.9     |     5.5     |
> | backchannel |  20.2 |   0.028   |     6.0     |     1.7     |
> |   welcome   |  17.6 |   0.025   |     6.4     |     2.7     |
> |  recommend  |  22.0 |   0.031   |     5.7     |     8.3     |
> |   nooffer   |  21.5 |   0.032   |     5.8     |     8.0     |
> |     book    |  21.3 |   0.034   |     5.9     |     7.4     |
> |    nobook   |  20.1 |   0.034   |     5.7     |     7.8     |
> |     edit    |  21.5 |   0.031   |     5.2     |     2.5     |
>
> Here is a summary of the observations:
>
>      a. Pitch:'inform' has the highest pitch, while 'bye' has the lowest. Most dialog acts have pitch values clustered around the range of 20 to 22.
>
>      b. Intensity (RMSE): 'inform' and 'request' show slightly higher RMSE values compared to others. The majority of the dialog acts have RMSE values clustered closely around the range of 0.03 to 0.035.
>
>      c. Speech Rate: 'reqmore' and 'welcome' exhibit the highest speech rate, while 'edit' shows the lowest. The speech rates for most dialog acts are in the range of 5.5 to 6.5.
>
>      d. Average Duration (in seconds): 'backchannel' has the shortest duration, and 'recommend' the longest. Dialog acts such as 'select', 'nooffer', and 'nobook' also have longer average durations compared to others like 'greet' and 'thanks'. From these observations, it is clear that different dialog acts have varying characteristics in terms of their audio features. This can be attributed to the inherent nature of the dialog act; for instance, 'recommend' or 'nooffer' might typically have more content and thus a longer duration, while backchannel or greet would be shorter.
>    Our current statistical information focuses on text data, and we will subsequently add additional statistical information on audio data in the Appendix. Thanks for the suggestion.
>
> Thanks for your suggestions for our paper, we believe that the revised version will be much improved. We also hope that our rebuttal can address your concerns.

---

### Official Review · Reviewer_QaYG · 2023-07-21
**This dataset has been dearly missed**

**Rating:** 9
**Confidence:** 4

**Strengths:**

It is a very good paper. I like everything about it. The dataset is very large (when compared to other annotated datasets used to train ASRs). The idea to put challenging annotations that test models' abilities to discover cross-turn entities, or to perform various types of reasoning on spontaneous speech is very good. I like the strict protocol used to generate conversations and to test the quality of recordings, transcriptions, and annotations. The fact that such significant work has been undertaken in the field of spontaneous speech is very important.

**Additional Feedback:**

This is an excellent resource. The area of spontaneous speech and spoken language understanding has been long blocked by the lack of sufficient training resources. The dataset presented in the publication is very good and very useful. My review is quite short, but I really don't have anything else to say except for "great job!"

**Clarity:**

The paper is very well written, clear and informative. The procedure of dataset compilation is described in detail, as well as the annotation for different language understanding tasks. The only thing I did not catch from the paper and the appendix is the number of annotations aiming at testing particular tasks, such as cross-turn slot detection or temporal reasoning.

**Correctness:**

The paper is complete and seems to be correct. The leaderboard is intuitive (one would expect that dual-mode training with alighment would produce the best results). One may argue if the metric used to construct the leaderboard is the best possible (it is essentially a custom formula combining three basic metrics representing entity retrieval, slot filling, and overall language quality). It might be a good idea to allow for more nuanced metrics or to introduce more metrics that would evaluate multiple criteria of models' answers.

**Documentation:**

The website contains the leaderboard and the links to download the audio and the transcriptions. The Datasheet for SpokenWOZ contains the purpose of dataset curation and the description of intended uses of the dataset. All the required details are provided in the auxiliary material accompanying the paper.

**Ethics:**

I don't find any issues around the design and the creation of the dataset that might create any type of ethical risks.

**Limitations:**

I don't see any limitations or flaws in the paper. It is sound, interesting, the effort to prepare the dataset was significant, there is really nothing I can point out as a drawback.

**Opportunities For Improvement:**

I don't think it would be fair to suggest any improvements to this work. I believe it absolutely merits the presentation at NeurIPS, the dataset is very useful and will be welcomed by the research community with enthusiasm. The website of the project is nice, with descriptions, figures, and the leaderboard. Maybe the only thing worth mentioning to the Authors (although I am sure they are already thinking about it) is the inclusion of newer generative models, and testing more advanced prompting techniques with LLMs - the paper examines only two such models and the results of the evaluation are really interesting).

**Relation To Prior Work:**

The discussion of the related work is quite short, but the domain of the dataset is quite specific (task-oriented audio recordings with transcriptions) and to the best of my knowledge, the Authors cover all available datasets. There is no discussion on general-purpose datasets for ASR training (such as CommonVoice, Fisher, CallHome, LibriSpeech, etc.), but these datasets are scripted (i.e. the conversations are not free) and few could quality as task-oriented. Maybe the Authors would like to include a single reference to a collection of such datasets for completeness sake.

**Summary And Contributions:**

The Authors create a very large dataset of transcripts and audio recordings of spontaneous dialogues that represent human-to-human conversations aimed at realizing a particular task (booking, buying, asking, ordering, etc.) The main contribution is the dataset itself, an extremely useful and high-quality resource that will help tremendously in developing modern conversational interfaces, voicebots, etc. The way the dataset has been designed and prepared is really impressive.

---

> ### Author Response · Authors · 2023-08-21
>
> We feel very excited to receive such a high rating from you for our paper. It makes us feel that our efforts are worthwhile! We hope that SpokenWOZ will contribute to the research community.
>
> Due to the limitation of the page of the paper, we do not discuss general-purpose datasets for ASR training (such as CommonVoice, Fisher, CallHome, LibriSpeech, etc.), we will add the above content in the revised version, thanks for pointing this out!

---

### Official Review · Reviewer_fz7a · 2023-07-28
**Long needed, solid and sizable speech Task-Oriented Dialogue dataset**

**Rating:** 9
**Confidence:** 4
**Correctness:** Yes - claims are correct and SpokenWO…
**Clarity:** Yes, this paper is well written.

**Strengths:**

The main strength of the submission is sharing with the research community ready to use, free, large dataset that costs to make around 55k according to estimates (excluding researchers' time).
What is another contribution is providing insights into creation processes making it extensible and reproducible. Moreover, it is thoroughly documented.
Next, introducing new slot (cross-turn and reasoning) types and proposing Macro Average Mentioned Slot Accuracy and using dual-model is noteworthy. Nonetheless demonstrated results and conclusions from experiments were to be expected. The generative method's helpfulness and dual-models superiority is not a surprising conclusion.
Achievement in itself is by means of reporting experiment results on SpokenWOZ (for both dialogue state tracking and response generation), successfully demonstrating that the SpokenWOZ is much more challenging than for example MultiWOZ - its properties stemming from its spoken nature are as planned posing issues for the vast majority of models performing well on written datasets.
It is great to see exposing shortcomings of LLMs, identifying what they are and tackling them.
Submission with primarily supplementary material proves beyond the shadow of a doubt the high quality of the research.
The ethical and social implications of a paper are positive. The research community will gain great materials to experiment with and build upon.


**Additional Feedback:**

It greatly contributes to the research community, with no additional feedback other than provided in the section “Opportunities for improvement”.

**Documentation:**

It has vast supplementary materials and documentation. They demonstrate that standard good practices were followed. Steps taken to create it, spending details were shared as well as steps of data collection, planned maintenance and datasets analysis / insights were documented.

**Ethics:**

Not really, I don’t

**Limitations:**

There are no negative societal impacts or limitations that should be addressed.

**Opportunities For Improvement:**

There are minor opportunities for improvement  - apart from the presence of gold standard transcripts also checking out other ASR systems on even a small subset of data due to cost constraints would be a valuable part of the study and showcase how ASR choice influences the rest of the pipeline.
More varied, perhaps paraphrased conversation templates with more realistic data could be beneficial too and would make experiments / models more robust.


**Relation To Prior Work:**

Yes, a total of 9 various models were assessed as well multiple written and spoken TOD datasets were discussed - their sizes, covered domains and tasks. The presence of audio on which ASR was run as well as being close to a real-life setting thanks to conversations being human-to-human make SpokenWOZ distinctive, but its scale is what makes it stand out (however not by a large margin / order of magnitude).

**Summary And Contributions:**

The paper presents the rationale, way of creation and extensive characteristics of the newly introduced SpokenWOZ - speech-text dataset for spoken task-oriented dialogue (TOD) with a total of 5.7k dialogues, including audio files and texts with metadata, documentation and meticulous description.
Moreover, it introduces cross-turn slot and reasoning slot (temporal / mathematical / semantic) detection - that is uncommon in written dialogues and what addresses the challenges that research community encounters while working with spoken dialogues.
It discusses conducted experiments with dialogue state tracking and response generation on SpokenWOZ, focusing the attention of the research community on the unique challenges of spoken dialogues and showcases that in spite of high hopes LLMs didn’t solve all NLP problems as of yet and dedicated dual-models are the way to go.

---

> ### Author Response · Authors · 2023-08-21
>
> Thank you very much for your kind words about our paper, and especially for considering SpokenWOZ as a great contribution to the research community!
>
> Regarding your concerns:
>
> 1. Indeed, due to cost constraints, we do not explore the impact of different ASR tools and different levels of ASR noise yet. It can be very interesting and we list the task as our future work.
>
> 2. It is also valuable and interesting to attract more researchers to conduct this experiment on SpokenWOZ.
>
> Finally, thank you again for your careful review of our paper!

---

### Official Review · Reviewer_F6kt · 2023-07-29
**Review for SpokenWOZ**

**Rating:** 5
**Confidence:** 4
**Correctness:** The evaluation metrics could be optim…
**Clarity:** The paper can improve the writing qua…

**Strengths:**

1. It contributes a large-scale speed-text TOD dataset across 8 domains and 249 hours, which is a valuable contribution to the dialogue research community.
2. It includes detailed analysis of the dataset and conducts quality control to improve the annotation quality of the dataset.


**Additional Feedback:**

1. It would be more beneficial to provide some generated responses of different models.
2. Referring to Table 15, the dialogue context merely contains "lavelle lodge". In this situation, it would likely be challenging for a model, and even a human, to provide the correct annotation as "lovell lodge" without referencing a database. The JGA used in [2] maybe not quite suitable as it strictly measures the string match. Is fuzzy match JGA better here?

**Documentation:**

Yes

**Limitations:**

Yes

**Opportunities For Improvement:**

1. While the importance of annotation quality in dialogue datasets is emphasized, this paper provides limited details on the methods used for quality control. The following questions remain:
    - What criteria were used to identify instances of missing annotations?
    - Given that users may express similar thoughts in a multitude of ways during a dialogue conversation (e.g., "14pm", "2pm in the afternoon" or "I need a parking", "Parking is yes"), how does the dataset link such diverse utterances to their corresponding ground truths?
    - The final utterance referenced in Section 3.2 appears to be outdated, as there are newer versions of MultiWOZ that notably enhance the overall dataset quality.
2. Including slot accuracy in the experiments could offer valuable additional insights into the model's performance.
3. The metrics currently used for evaluating policy optimization and end-to-end modeling may not be fully applicable to Large Language Models (LLMs). The INFORM and SUCCESS metrics from MultiWOZ seem to be designed primarily for de-lexicalized responses rather than realistic generation. Evaluating LLMs might be more effective if the focus is on lexicalized responses - that is, responses containing actual slot values rather than placeholders.







**Relation To Prior Work:**

Yes

**Summary And Contributions:**

This paper introduces SpokenWOZ, a large-scale speech-text Task-Oriented Dialogue (TOD) dataset. It contains more than 5.7k dialogues and 249 hours of audio. The authors provide an in-depth analysis of this dataset and propose two new tasks. By conducting experiments on established TOD baselines and Large Language Models (LLMs), the paper highlights the challenges presented by this dataset.

---

> ### Author Response · Authors · 2023-08-21
>
> Thanks for your thoughtful reviews. We are encouraged that you consider SpokenWOZ a valuable contribution to the research community.
>
> In response to your concerns:
>
> 1. More details about quality control are provided below:
>
>      a. We identify instances of missing annotations by comparing the task goal provided to participants with the dialogue state of the last turn in that dialogue, and if the content in the task goal is not fully covered in the annotated dialogue state, we consider that there is the instance of missing annotation.
>
>      b. During the annotation, such as annotating the dialogue state, we will annotate the value according to the designed ontology. For example, "14pm", "2pm in the afternoon" will be annotated as "time: 14:00", and "I need a parking", "Parking is yes" will be labeled as "parking: yes". We also provide the corresponding ontology description file in the data.
>
>      c. It is great suggestion to evaluate the annotation performance in the newer version of MultiWOZ. Due to time limits, we randomly sample 100 turns from newer version namely MultiWOZ 2.3 to check whether their dialogue act and dialogue state are correctly annotated. By our manual annotation, we find that only 87% of the sampled turns are annotated exactly correctly, which is much lower than that of the SpokenWOZ. Meanwhile, we do not compare the latest version of MultiWOZ 2.4 because it does not include the annotation of the dialogue act.
>
>      d. We will add these details in the revised version, thanks for pointing them out.
>
> 2. Thanks for your constructive comments, we also consider that slot accuracy can offer additional insights. However, the values of most of the slots in the dialogue correspond to values are “null”, the slot accuracy tends to be high. To improve this metric, we introduce Macro Average Mentioned Slot Accuracy (MAMS) to measure the difficulty of different types of slots in DST. In this scenario, MAMS is a better metric than the slot accuracy measurement.
>
> 3. For your concerns about INFORM and SUCCESS:
>
>      a. As detailed in Appendix A.9.1, We used the following approach to evaluate the ability of LLMs to generate responses: We first use the dialogue state to query the provided database. Then we use the contents of the database and the generated lexicalized response to match the strings and complete the de-lexicalization process. The de-lexicalized response will be used to calculate INFORM and SUCCESS. In this way, LLMs do not directly generate de-lexicalized responses, so INFORM and SUCCESS do not underestimate LLM's performance.
>
>      b. For supervised models, delexicalization can replace slot values with their corresponding slot tokens and allow these models to learn value-independent parameters. This will underestimate the performance of supervised models if de-lexicalization is not used.
>
>      c. In order to fairly compare supervised models and LLMs, we use the above methods as well as the two metrics INFORM and SUCCESS to evaluate their performance.
>
>      d. Also, using de-lexicalized responses for evaluation, rather than simply comparing generated responses to lexicalized ground truth responses, allows a fairer evaluation of situations when there are more than one entity that satisfies the goal constraints in the database.
>
> For the Additional Feedback:
>
> 1. As shown in Appendix A.9.3, we list the generated response from ChatGPT. Here are some generated response form ChatGPT and supervised SPACE+WavLM_{align} in Policy Optimization task. The supervised model can give the shorter response, which is more in line with the spoken conversation, while the response form ChatGPT is much longer.
>
>     ○ USER: um yes, I'm looking for a particular restaurant in Cambridge.
>
>     ○ ChatGPT: Of course! I'd be happy to help you find a restaurant in Cambridge. Could you please provide me with more details about the restaurant you're looking for? Specifically, do you have a name or any specific preferences like the type of cuisine or price range you're interested in?
>
>     ○ SPACE+WavLM_{align}: okay, sure. May I know more details, please.
>
>     ○ Ground Truth: Yes. May I have the name please?
>
>      We will provide generated responses of different supervised models in the revised version.
>
> 2. Thanks for pointing this out. We think that the case you mentioned exactly reflects the new challenges posed by ASR errors, which make the JGA metrics look very strict because current models do not pay attention to the ASR problem. Meanwhile, as shown in Appendix A.3, we find that supervised generative models can be somewhat robust to the noisy user utterances. Subsequently, we will update the leaderboard with new metrics proposed by other researchers to evaluate the existing models.
>
> Thank you for your suggestions for our paper, we believe that the revised version will be more clear and sound. We also hope that you can kindly increase your score if our response has helped address your concerns.

---

### Official Review · Reviewer_cbUA · 2023-08-03
**Good paper**

**Rating:** 7
**Confidence:** 4
**Correctness:** n/a
**Clarity:** n/a

**Strengths:**

1. The dataset contains 8 domains, 203k turns, 5.7k dialogues and 249 hours of audios.
2. The datasets addresses some unique characteristics of spoken conversation.

**Additional Feedback:**

n/a

**Documentation:**

n/a

**Opportunities For Improvement:**

1. In section 3.2,  It is mentioned that the dialogue state and dialogue act are annotated using text transcriptions generated by the ASR tool. why using ASR tool? I think there should be ASR noise. Why not let the annotators first do the job of ASR?
2. The impact of different levels ASR noise should be studies.
3. The writing can be improved (especially for the problem of tense confusion).


**Relation To Prior Work:**

n/a

**Summary And Contributions:**

This paper presents a large-scale spoken speech-text dataset for spoken TOD. It captures the unique characteristics of spoken conversation (e.g., word-by-word processing and reasoning in spoken language) that were ignored in previous study. Various baselines are tested on the newly proposed datasets. The results suggest that there is  substantial room for improvement.

---

> ### Author Response · Authors · 2023-08-21
>
> Thanks for your constructive reviews. It is encouraging to see your positive comments.
>
> Regarding to your concerns:
> 1.  We apologize that our writing may lead to misunderstanding, and we will explain our methods more clearly below:
>
>      a. In fact, we use ASR transcriptions in order to improve annotation efficiency and reduce annotation time. During the annotation process of dialogue state and dialogue act of user utterances, the annotators annotate them based on both audio and text transcriptions generated by the ASR tool. Meanwhile, the annotators need to fill in the normalized, ASR noise-free annotation values according to the designed ontology. In this way, we can get the correct annotation content instead of the noisy one.
>
>      b. When annotating the position of the dialogue act in the user utterance, annotators annotate the start and end positions in the noisy utterance. In this way, we can keep the real-world ASR noise in the user utterance and the clean annotation.
>
>      c. For agent utterances, annotators need to transcribe the audio to obtain the clean text and then annotate dialogue state and dialogue act given the clean text.
>
>      d. At the same time, we use the ASR tool in order to make the real-world ASR noise preserved in the user utterance.
>
>     Accordingly, we do not have the raised issue of the ASR noise in annotations.
>
> 2. Thank you for your constructive comments. We also consider this exploration very interesting. However, we do not explore the impact of different levels of ASR noise because of the following considerations:
>
>      a. We can use different ASR tools as part of the pipeline to explore the impact of different levels ASR noise on subsequent modules in the pipeline. However, when annotating the position of dialogue act of user utterances, different ASR transcripted utterances would require new corresponding annotations, which are used in response generation tasks.
>
>      b. Multiple repetitive annotations and the use of different ASR tools would result in additional overhead. Due to the cost constraints, we need new budget to run this experiment and hence we list it as our future work. ​
>
> 3. We will improve our writing in our revised version. Thanks for your suggestions to make our paper stronger and more solid.

---

### Decision · Program_Chairs · 2023-09-22

**Decision:**

Accept (Poster)

**Comment:**

This paper presents a new spoken, multi-domain, task-oriented dialogue dataset, that closely parallels the widely used MultiWOZ multi-domain task-oriented dataset. Reviewers included good reviews, mainly suggestions for including clarifications in the paper, and the authors positively responded to these suggestions. I agree with the reviewers that this dataset will be useful.
Some additional concerns include: whether the annotation of the ASR outputs caused any noise in the annotations, especially for dialogue state categories. Please clarify this in the paper.
Furthermore, it would have been better if the authors had extended the databases and locales for this collection, as the limited database has been a concern for MultiWOZ. I hope the authors can consider this in the future updates.